# ROTATIVE FACTORIZATION MACHINES

## ABSTRACT

Feature interaction learning, which focuses on capturing the complex relationships among multiple features, is crucial in various real-world predictive tasks. However, most feature interaction approaches empirically enumerate all feature interactions within a predefined maximal order, which leads to suboptimal results due to the restricted learning capacity. Some recent studies propose intricate transformations to convert the feature interaction orders into learnable parameters, enabling them to automatically learn the interactions from data. Despite the progress, the interaction order of each feature is often independently learned, which lacks the flexibility to capture the feature dependencies in the varying context. In addition, they can only model the feature interactions within a bounded order due to the exponential growth of the interaction terms. To address these issues, we present a Rotative Factorization Machine (**RFM**). Unlike prior studies, RFM represents each feature as a polar angle in the complex plane. As such, the feature interactions are converted into a series of complex rotations, where the orders are cast into the rotation coefficients, thereby allowing for the learning of arbitrarily large order. Further, we propose a novel self-attentive rotation function that models the rotation coefficients through a rotation-based attention mechanism, which can adaptively learn the interaction orders from different interaction contexts. Moreover, it incorporates a modulus amplification network to learn the modulus of the complex features that further enhances the representations. Such a network can adaptively capture the feature interactions in the varying context, with no need of predefined order coefficients. Extensive experiments conducted on five widely used datasets have demonstrated the effectiveness of our approach.

## 1 INTRODUCTION

Feature interaction learning is crucial for the success of various real-world predictive tasks, such as click-through rate (CTR) predictions and product recommendations. The key to learning effective feature interactions is to accurately model the complex relationship among multiple features. Typically, a feature interaction term is modeled as a combination of input features with their respective *interaction orders*, formally denoted by $e_1^{\alpha_1} \odot \cdots \odot e_m^{\alpha_m}$. The order $\alpha_j$ determines the effect of the $j$-th feature and $\alpha_j = 0$ discards the corresponding feature $e_j$. In the literature, various methods have been proposed for learning effective feature interactions, from early factorization machines (*e.g.,* FM (Rendle, 2010)) to recent deep neural networks (*e.g.,* CrossNet (Wang et al., 2021)).

Typically, existing methods have adopted a similar modeling approach: they often set a maximal order, and consider conducting feature interactions within the predefined order. Despite the progress, they suffer from a decline in model capability owing to the suboptimal learning of the restricted orders (*e.g., integer-only order* (Lian et al., 2018)). Further, due to the exponential growth of feature combinations, they can only learn the interactions within a small order to maintain efficiency, *e.g.,* FM (Rendle, 2010) only considers second-order feature interactions.

Considering the above limitations, several studies (Cheng et al., 2020; Tian et al., 2023; Cai et al., 2021) propose to automatically learn the interaction orders from data. The core idea of these approaches is to map features into a special vector space (*e.g.,* logarithmic vector space (Cheng et al., 2020)). As such, the exponential form of interaction terms (*i.e.,* $\prod e_j^{\alpha_j}$) is converted to linear combinations (*i.e.,* $\exp\left(\sum \alpha_j \log e_j\right)$), and the orders (*i.e.,* $\alpha_j$) are cast into learnable linear coefficients, allowing for the learning of adaptive-order interactions. Generally, existing methods learn the orders either in a *field-aware* way or in an *instance-aware* way. As shown in Figure 1(a) and Figure 1(b),

given two fields along with their feature interaction, field-aware methods learn a shared order for all features from the same field (*e.g.,* $\alpha_G$ is shared by $\mathrm{Male}, \mathrm{Female}$ for field $\mathrm{Gender}$), capturing the field-level importance, whereas the instance-aware methods assign a specific order for each feature (*e.g.,* $\alpha_M, \alpha_F$ for $\mathrm{Male}, \mathrm{Female}$) to learn the feature importance.

Although these approaches are capable of capturing the underlying relationships in real-world scenarios, they still have two limitations. First, the interaction order of each feature is independently learned, which lacks the flexibility to capture the *feature dependencies* in the varying context. As increasing evidence shows (Wang et al., 2022), in real-world applications, the importance of a certain feature is often influenced by other features. For example, considering the feature interaction $\langle \mathrm{UserGender}, \mathrm{MovieGenre}, \mathrm{Actor} \rangle$ in the scenario of movie recommendations, the Actor features may have varying effects for Idol and Horror movie genres. However, it is challenging for both field-aware and instance-aware models to effectively capture such varied feature importance in different interaction contexts. As such, we argue that the importance of a specific feature should be adaptively learned depending on the other features it is involved with, which is called *relation-aware* (See Figure 1(c)) in this paper. Second, since the interaction terms exponentially grow with the order, these methods often model the interactions within a *bounded order*[1], which cannot scale to the high-order cases in industrial scenarios. Considering these limitations, we aim to seek a more effective approach to adaptively learn the interaction order in a *relation-aware* way, meanwhile surpass the scale limits of interaction order in existing work.

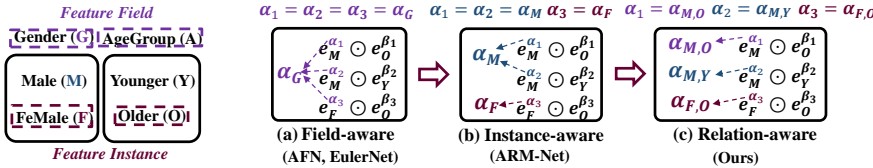

Figure 1: Comparisons of three feature interaction approaches. Field-aware methods set a fixed interaction order for each *feature field*; instance-aware methods set a unique interaction order for each *feature instance* (*a.k.a.,* feature value); relation-aware methods set a unique interaction order for each *feature combination*.

To this end, this paper presents a novel rotative factorization machine (**RFM**), for adaptively learning the *unbounded-order* feature interactions in a *relation-aware* way. Unlike prior work, the key idea of RFM is to represent each feature as a *polar angle* (*i.e.,* $e^{i\boldsymbol{\theta}_j}$) in the complex plane, and conduct the *attentive rotations* to model complicated feature interactions. For learning the *unbounded-order* feature interactions, RFM converts the feature interactions into the *complex rotations* (*i.e.,* $\exp(i \sum \alpha_j \boldsymbol{\theta}_j)$), where the interaction orders are cast into the *rotation coefficients* (*i.e.,* $\alpha_j$), thereby avoiding the exponential explosion of the interaction terms. For learning the feature interactions in a *relation-aware* way, we propose a novel self-attentive rotation function (*i.e.,* $\exp(i \sum \alpha_{j,l} \boldsymbol{\theta}_l)$), where the rotation coefficients (*i.e.,* $\alpha_{j,l}$) are learned by a *rotation-based* attention mechanism, capturing the dependencies between feature $j$ and $l$. Moreover, we devise a modulus amplification network to learn the modulus of the complex features that further enhances the feature interaction learning. Such a network can model all three types of feature interaction patterns (*i.e.,* *field-aware*, *instance-aware* and *relation-aware*), with no need of pre-specified order coefficients.

To our knowledge, it is the first work that is capable of learning the interactions with arbitrarily large order adaptively from the corresponding interaction contexts. Furthermore, it has been proven that our approach can be instantiated to a variety of traditional inner-product based interaction models (*e.g.,* FM (Rendle, 2010)). To evaluate our model, we conduct extensive experiments on five public datasets, and the experimental results show that our model consistently outperforms a number of competitive feature interaction approaches.

## 2 Preliminary

As the key technique in many prediction tasks (Zhang et al., 2021; Xiao & Benbasat, 2007), feature interaction modeling aims to capture the underlying relationships among multiple features. It takes as input a concatenated vector of features, denoted as $\boldsymbol{x} = [\boldsymbol{x}_1, ..., \boldsymbol{x}_m]$, where $m$ represents the number of feature fields (*e.g., Gender*), and $\boldsymbol{x}_j$ is the one-hot vector of a feature instance (*e.g.,*

---

[1]Due to gradient explosion, they cannot learn a large interaction order (*e.g.,* $\geq 70$, See Section 4.3).

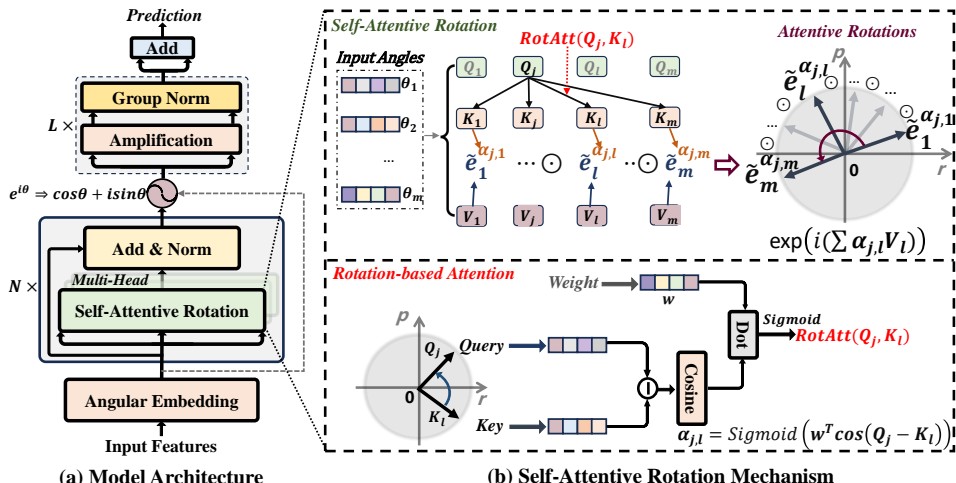

Figure 2: Architecture and components of our proposed rotative factorization machines.

*Male*) in the $j$-th field. Due to the high-dimensional, sparse nature of $x$, an embedding look-up operation $\mathrm{E}(\cdot)$ is often used to map each feature into a $d$-dimensional vector $e_j = \mathrm{E}(x_j) \in \mathbb{R}^d$. In this context, the feature interaction learning function $\mathcal{F}(\cdot)$ is commonly defined as:

$$\mathcal{F}(\mathcal{A}) = \sum_{\alpha \in \mathcal{A}} e_1^{\alpha_1} \odot e_2^{\alpha_2} \odot \cdots \odot e_m^{\alpha_m}, \qquad (1)$$

where $\odot$ denotes the element-wise product, $\mathcal{A}$ represents the set of all interaction orders, and each $\alpha \in \mathcal{A}$ specifies the order for each feature. While many methods manually set feature interaction orders, for instance, FM (Rendle, 2010) assigns $\mathcal{A} = \{\alpha | \sum_{j=1}^{m} \alpha_j = 2, \forall \alpha_j \in \{0, 1\}\}$ to capture second-order feature interactions. Further, AFN (Cheng et al., 2020) and EulerNet (Tian et al., 2023) propose automatically learning orders (*i.e.,* $\mathcal{A}$) from data. However, they primarily capture *field-aware* interactions, where the order $\alpha$ is shared across all features within a field. ARM-Net (Cai et al., 2021), as a promising approach, introduces a gated attention $\mathrm{Gate}(\cdot)$ for *instance-aware* interactions, with $\alpha_j = \mathrm{Gate}(e_j)$ evaluating feature importance. In contrast, we focus on learning *relation-aware* interactions, where $\alpha_j$ considers the dependencies between $e_j$ and other features.

## 3 METHODOLOGY

In this section, we present the proposed **R**otative **F**actorization **M**achines (**RFM**) (Figure 2(a)) for better modeling feature interactions in the prediction tasks. Unlike prior work, we represent each feature as a *polar angle* in the complex plane and use the *attentive rotations* to model complicated feature interactions. Specially, we focus on adaptively learning the *unbounded-order* feature interactions in a *relation-aware* way. For learning *unbounded-order* interactions, we convert the interactions into the *complex rotations* that casts the orders into the *rotation coefficients*, allowing for the learning of arbitrarily large order. For learning *relation-aware* interactions, we propose a self-attentive rotation layer, which can adaptively learn the orders from different interaction contexts. Moreover, a modulus amplification network is incorporated to learn the modulus of the complex features for enhancing the representations. In what follows, we introduce the details of relation-aware interaction modeling (Section 3.1) and the modulus amplification network (Section 3.2).

### 3.1 RELATION-AWARE FEATURE INTERACTION LEARNING

As discussed in Section 1, prior work mainly learns the feature interactions in a *field-aware* or *instance-aware* way (directly optimizing Eq.1), suffering from two major limitations. First, since the term $e_j^{\alpha_j}$ exponentially grows with the power $\alpha_j$, they can only learn the interactions within a *bounded order*, which cannot scale to the high-order cases. Second, the interaction order of each feature is independently learned. It is difficult for them to learn the feature dependencies in the varying context that leads to the suboptimal performance. To address these issues, we represent each feature as a polar angle in the complex plane and propose a self-attentive rotation layer for learning the *relation-aware* feature interactions.

### 3.1.1 ANGULAR REPRESENTATION OF FEATURES

As mentioned in Section 2, the feature $\boldsymbol{x}_j$ can be mapped into a vector embedding via the look-up operation $\mathrm{E}(\cdot)$. Due to the exponential explosion, it is challenging to effectively learn high-order interactions. Our solution is to represent the features as a set of *polar angles* in the complex plane:

$$\boldsymbol{\theta}_j = \mathrm{E}(\boldsymbol{x}_j), \;\; \tilde{\boldsymbol{e}}_j = e^{i\boldsymbol{\theta}_j}, \tag{2}$$

where $i$ is the imaginary unit that satisfies $i^2 = -1$. In this way, given the angular feature representations $\{\tilde{\boldsymbol{e}}_j\}_{j=1}^m \in \mathbb{C}^{m \times d}$, the interactions are cast into a series of *complex rotations*:

$$\mathcal{F}(\mathcal{A}) = \sum_{\boldsymbol{\alpha} \in \mathcal{A}} \tilde{\boldsymbol{e}}_1^{\alpha_1} \odot \tilde{\boldsymbol{e}}_2^{\alpha_2} \odot \cdots \odot \tilde{\boldsymbol{e}}_m^{\alpha_m} = \sum_{\boldsymbol{\alpha} \in \mathcal{A}} \underbrace{\exp\left(i \sum_{j=1}^m \alpha_j \boldsymbol{\theta}_j\right)}_{\text{Complex Rotation}}. \tag{3}$$

In mathematics, a *complex rotation* (*i.e.*, $\exp(i \sum_{j=1}^m \alpha_j \boldsymbol{\theta}_j)$) performs a linear transformation on the phase of the complex vectors without affecting their modulus. In our case, we utilize it to model the complicated interactions, and use the *rotation coefficients* (*i.e.*, $\alpha_j$) to model the interaction orders. As such, the interactions are learned on a *unit circle* (*i.e.*, modulus are fixed to 1) with a finite norm:

$$\|\mathcal{F}(\mathcal{A})\| = \left\| \sum_{\boldsymbol{\alpha} \in \mathcal{A}} \exp\left(i \sum_{j=1}^m \alpha_j \boldsymbol{\theta}_j\right) \right\| \leq \sum_{\boldsymbol{\alpha} \in \mathcal{A}} \left\| \exp\left(i \sum_{j=1}^m \alpha_j \boldsymbol{\theta}_j\right) \right\| \leq |\mathcal{A}|d. \tag{4}$$

Since the upper bound is independent of the order $\alpha_j$, it can effectively learn complicated interactions with arbitrarily large order, without limitations in prior work (*e.g., exponential explosion*).

### 3.1.2 SELF-ATTENTIVE ROTATION

The self-attentive rotation layer is the core of our proposed RFM for learning the *relation-aware* feature interactions. As shown in Figure 2(b), the key idea of this layer is to conduct the *attentive rotations* with the rotation coefficients modeled by a *rotation-based attention* mechanism, thereby allowing for the adaptive learning of feature dependencies in the varying context. As such, it takes as input a set of angles and outputs a set of rotated angles, and thus we can stack multiple such layers to form a capable network. Here we describe the attentive rotations within a single layer.

**Rotation-based Attention for Attentive Rotations.** As shown in Eq. 3, the interaction with the order $\boldsymbol{\alpha}$ is cast into a complex rotation (*i.e.*, $\exp(i \sum_{j=1}^m \alpha_j \boldsymbol{\theta}_j)$). To learn the *relation-aware* interactions, a major issue is how to effectively model the feature dependencies in the varying context. Typically, the self-attention mechanism (Vaswani et al., 2017) has shown excellent capacity in modeling complicated dependencies. However, it is designed to model relationships for real vectors, which is not suitable for modeling relationships among angular representations. As our solution, we propose a rotation-based attention mechanism to adaptively model the rotation coefficients (*i.e.*, $\alpha_j$), which enables it to effectively learn the dependencies between different angle-represented features.

As shown in Figure 2(b), we adopt a key-value based self-attention to conduct the attentive rotations. Specifically, the query-key pairs with similar angles are considered more important. Given the input $\{\boldsymbol{\theta}_j\}_{j=1}^m$, the dependency between feature $j$ and $l$ is learned by the rotation angle from key to query:

$$\alpha_{j,l} = \mathrm{RotAtt}(\boldsymbol{Q}_j, \boldsymbol{K}_l) = \mathrm{Sigmoid}(\boldsymbol{w}^\top \cos(\boldsymbol{\theta}_j^Q - \boldsymbol{\theta}_l^K)), \tag{5}$$

$$\boldsymbol{Q}_j^\top = \boldsymbol{\theta}_j^Q = \boldsymbol{W}_j^Q \boldsymbol{\theta}_j, \;\; \boldsymbol{K}_l^\top = \boldsymbol{\theta}_l^K = \boldsymbol{W}_l^K \boldsymbol{\theta}_l, \tag{6}$$

where $\boldsymbol{w} \in \mathbb{R}^d$ is a weight vector. To improve the field-specific semantics, we utilize a set of field-specific matrices $\{\boldsymbol{W}_j^Q \in \mathbb{R}^{d' \times d}\}_{j=1}^m, \{\boldsymbol{W}_l^K \in \mathbb{R}^{d' \times d}\}_{l=1}^m$ to map the features into a set of queries $\{\boldsymbol{\theta}_j^Q\}_{j=1}^m$ and keys $\{\boldsymbol{\theta}_l^K\}_{l=1}^m$, and pack them together into two matrices $\boldsymbol{Q}, \boldsymbol{K} \in \mathbb{R}^{m \times d'}$. Likewise, the values are also packed into matrix $\boldsymbol{V} \in \mathbb{R}^{m \times d'}$. Further, we aggregate all contextual information of feature $j$ as $\tilde{\boldsymbol{\theta}}_j = \sum_{l=1}^m \alpha_{j,l} \boldsymbol{\theta}_l^V$. As such, the interaction with order $\mathcal{A}_j = \{\boldsymbol{\alpha}_j\}$ is cast into a *self-attentive rotation* with coefficients learned by the proposed rotation-based attention (Eq. 5):

$$\mathcal{F}(\mathcal{A}_j) = \exp\underbrace{\left(i \sum_{l=1}^m \alpha_{j,l} \boldsymbol{\theta}_l^V\right)}_{\text{Self-Attentive Rotation}} = \exp(i\tilde{\boldsymbol{\theta}}_j). \tag{7}$$

This formula is the core of RFM for learning the *relation-aware* interactions. Different from prior work, the rotation coefficient $\alpha_{j,l}$, which also represents the interaction order, is learned through the self-attention mechanism, capturing the dependencies between feature $j$ and $l$. In practice, we pack the orders into a matrix $\boldsymbol{A}$ (*i.e.*, $\boldsymbol{A}_{j,l} = \alpha_{j,l}$), to aggregate the contextual information of all features:

$$\text{AttentiveRo}(\boldsymbol{Q}, \boldsymbol{K}, \boldsymbol{V}) = [\tilde{\boldsymbol{\theta}}_1, \tilde{\boldsymbol{\theta}}_2, \cdots, \tilde{\boldsymbol{\theta}}_m]^\top = \boldsymbol{A}\boldsymbol{V}. \tag{8}$$

Formally, the tensor-form calculation of the rotation-based attention score can be also given by:

$$\boldsymbol{A} = \text{RotAtt}(\boldsymbol{Q}, \boldsymbol{K}) = \text{Sigmoid}\Big(\text{Re}\Big[\Big(\exp(i\boldsymbol{Q})\text{diag}(\boldsymbol{w})\Big)\exp(-i\boldsymbol{K})^\top\Big]\Big), \tag{9}$$

where $\text{Re}[\cdot]$ returns the real part of a complex vector. *See proof in Appendix A.1.*

**Multi-Head Rotation.** To learn diversified contextual information from different subspaces, we extend RFM to adopt a multi-head rotation. Specifically, we introduce $h$ independent attention heads performing the rotation function of Eq. 8, and then concatenate them to obtain final representations:

$$\text{MultiHeadRo}(\boldsymbol{Q}, \boldsymbol{K}, \boldsymbol{V}) = \text{Concat}(\text{head}_1, \text{head}_2, ..., \text{head}_h), \tag{10}$$

$$\text{head}_j = \text{AttentiveRo}(\boldsymbol{Q}\boldsymbol{H}_j^Q, \boldsymbol{K}\boldsymbol{H}_j^K, \boldsymbol{V}\boldsymbol{H}_j^V), \tag{11}$$

where $\boldsymbol{H}_j^Q, \boldsymbol{H}_j^K, \boldsymbol{H}_j^V \in \mathbb{R}^{d' \times d_h}$ are projection matrices, $d_h = d'/h$. In this way, we can use the head number $h$ to control the number of feature interaction terms. Further, we can stack multiple layers by taking the output representations of the previous layer as the input for the next layer, and set varying $h$ at different layers to increase the model flexibility. Besides, to preserve the previously learned representations, we follow the transformer Miller et al. (2016) that employs a residual connection with a layer normalization (Ba et al., 2016) around each layer.

## 3.2 MODULUS AMPLIFICATION FOR ENHANCED FEATURE INTERACTION LEARNING

In the above rotation procedure, the features are limited to a unit circle with a fixed modulus of one, which may limit the model's capacity and lead to suboptimal results. For further enhancing the interaction learning, we devise a modulus amplification network to learn the modulus of the features.

**Coordinate Transformation.** For learning the modulus of the complex features, a straightforward approach is to feed them into a feed-forward neural network. However, it cannot effectively learn the representations since all features have the same modulus (*i.e.,* 1) after rotations. Instead of directly learning the modulus of the complex features, our solution is to optimize their real and imaginary parts. Specifically, given the output representation $e^{i\tilde{\boldsymbol{\theta}}_j}$ (See Eq. 8) of the last self-attentive rotation layer, we use the Euler's formula (*i.e.,* $e^{i\boldsymbol{\theta}} = \cos\boldsymbol{\theta} + i\sin\boldsymbol{\theta}$) to obtain its real and imaginary parts:

$$\boldsymbol{r}_j = \cos\tilde{\boldsymbol{\theta}}_j, \quad \boldsymbol{p}_j = \sin\tilde{\boldsymbol{\theta}}_j, \tag{12}$$

where $j \in \{1, ..., m\}$. After the coordinate transformation, each feature is represented by a rectangular-form complex vector, *i.e.,* $\boldsymbol{r}_j + i\boldsymbol{p}_j$. We utilize the complex representations $\{\boldsymbol{r}_j + i\boldsymbol{p}_j\}_{j=1}^m$ for the subsequent modulus amplification procedure. Further, we can optionally add a residual connection of the original (*i.e.,* first-order) features (See Eq. 2) to improve the low-order interactions.

**Modulus Amplification.** Given the representations in the rectangular form $\{\boldsymbol{r}_j + i\boldsymbol{p}_j\}_{j=1}^m$, we concatenate their real and imaginary parts and feed them into a shared multi-layer perception (MLP):

$$\boldsymbol{r}^{(0)} = \text{Concat}(\boldsymbol{r}_1, ..., \boldsymbol{r}_m), \qquad \boldsymbol{p}^{(0)} = \text{Concat}(\boldsymbol{p}_1, ..., \boldsymbol{p}_m), \tag{13}$$

$$\boldsymbol{r}^{(k)} = \text{GN}(\sigma(\boldsymbol{W}_k\boldsymbol{r}^{(k-1)} + \boldsymbol{b}_k)), \quad \boldsymbol{p}^{(k)} = \text{GN}(\sigma(\boldsymbol{W}_k\boldsymbol{p}^{(k-1)} + \boldsymbol{b}_k)), \tag{14}$$

where $k \in \{1, 2, ..., L\}$, $L$ is the depth, $\sigma$ is the activation function, $\boldsymbol{W}_k$ and $\boldsymbol{b}_k$ are the weight and bias of the $k$-th layer. In the above transformations, all feature vectors are concatenated into a long hidden vector as the input of the MLP, which may diminish the vector-based representation of each feature. To address this problem, we use the group normalization (Wu & He, 2018) $\text{GN}(\cdot)$ to preserve the feature-wise information. Formally, given the input vector $\boldsymbol{X} \in \mathbb{R}^D$, we view it as having $f$ latent features (*i.e.,* $\boldsymbol{X} = [\boldsymbol{X}_1, \boldsymbol{X}_2, ..., \boldsymbol{X}_f], f \mid D$), and $\text{GN}(\cdot)$ is formulated as follows:

$$\text{GN}(\boldsymbol{X}_j) = \gamma \cdot \frac{\boldsymbol{X}_j - \mu_j}{\sqrt{\sigma_j^2 + \epsilon}} + \beta, \tag{15}$$

where $j \in \{1, 2, ..., f\}$, $\mu_j$ and $\sigma_j$ denote the mean and standard deviation of $\boldsymbol{X}_j$, the scale parameter $\gamma$ and shift parameter $\beta$ are set to be trainable to enhance the representation of the GN$(\cdot)$ layer.

**Predictions for Model Training.** For predictions, we follow the prior work (Tian et al., 2023) that incorporates a transition weight $\boldsymbol{u}$ to project the representation of the last layer (*i.e.,* $\boldsymbol{r}^{(L)} + i\boldsymbol{p}^{(L)}$):

$$z = \boldsymbol{u}^\top (\boldsymbol{r}^{(L)} + i\boldsymbol{p}^{(L)}) = z_r + iz_p, \tag{16}$$

$$\hat{y} = \sigma(z_r + z_p). \tag{17}$$

Similar to FM (Rendle, 2010), RFM can be applied to a variety of tasks, such as classification and regression. Taking the binary classification tasks (*e.g.,* click-through rate prediction) for example, we use the widely-used binary cross-entropy loss with a regularization term to train our model:

$$\mathcal{L}(\boldsymbol{\Theta}) = -\frac{1}{N} \sum_{j=1}^{N} \left( y_j \log(\hat{y}_j) + (1 - y_j) \log(1 - \hat{y}_j) \right) + \lambda ||\boldsymbol{\Theta}||_2^2, \tag{18}$$

where $y_j$ and $\hat{y}_j$ are the ground-truth label and predicted result of $j$-th instance respectively, $\boldsymbol{\Theta}$ is the set of model parameters, and $\lambda$ is the $L_2$-norm penalty.

## 3.3 DISCUSSION

With the above transformations, RFM is able to model all three types of feature interaction patterns (*i.e., field-aware*, *instance-aware* and *relation-aware*) introduced in the Figure 1, meanwhile surpass the order limits in existing studies (See proofs in Eq. 4). Formally, we have the following finding:

**Theorem 3.1.** *If embeddings* $\{\boldsymbol{\theta}_j\}_{j=1}^m \in \mathbb{R}^{m \times d}$ *are* $L_2$*-regularized such that* $||\boldsymbol{\theta}_j||_2 \leq 1, \forall j \in \{1, ..., m\}$, *RFM can model the feature interaction pattern* $\boldsymbol{\Delta}_R = \boldsymbol{e}_1^{\alpha_{j,1}} \odot \boldsymbol{e}_2^{\alpha_{j,2}} \odot \cdots \odot \boldsymbol{e}_m^{\alpha_{j,m}}$, *with a probability of at least* $\mathcal{O}(1 - m/d)$ *that satisfies the maximum prediction error* $\mathcal{R} = \max(|\boldsymbol{\Delta}_{RFM} - \boldsymbol{\Delta}_R|) < \mathcal{O}(2 \sum_{k=1}^m \alpha_{j,k} \cdot \sqrt{\ln d/(d-1)})$. *Here* $\boldsymbol{e}_j \in \mathbb{R}^d$, $j \in \{1, ..., m\}$, $\alpha_{j,k} = f(\boldsymbol{e}_j, \boldsymbol{e}_k)$, *and* $f : \mathbb{R}^d \times \mathbb{R}^d \to \{0, 1\}$ *is any given feature dependency function. See proof in Appendix A.2.*

It indicates that in high-dimensional spaces, RFM can effectively learn the given feature relationships in real-world scenarios with infinitesimal loss. Further, the interactions learned in RFM can cover both the *field-aware* and *instance-aware* interactions (See proof in Appendix A.3). Specially, the inner product-based interactions (*e.g.,* FM (Rendle, 2010)) are special cases of our proposed rotation-based interactions (See Lemma A.1 and A.2). To our knowledge, RFM is the first work that proposes an attentive rotation mechanism for learning the *unbounded-order* interactions. In the literature, AFN (Cheng et al., 2020), EulerNet (Tian et al., 2023) and ARMNet (Cai et al., 2021) have also proposed to model the adaptive-order interactions, but the order of each feature is independently learned, which lacks the flexibility to capture the feature dependencies in the varying context. Although EulerNet has proposed to enhance the representations in the complex vector space, it still suffers from the exponential explosion issue when dealing with a large order (See Section 4.3), due to the exponential growth in the modulus of the complex features. In contrast, RFM is more *flexible, robust* in accurately learning the complicated feature interactions with arbitrarily large order involving massive feature fields. The comparison of these approaches is presented in Table 1.

Table 1: Comparison of different methods.

| Methods | Adaptive Order | Unbounded Order | Interaction Type |
|---|---|---|---|
| FM | ✗ | ✗ | Field |
| AFN | ✓ | ✗ | Field |
| ARM-Net | ✓ | ✗ | Field, Instance |
| EulerNet | ✓ | ✗ | Field |
| RFM | ✓ | ✓ | Field, Instance, Relation |

Table 2: Statistics of all datasets.

| Datasets | #Field | #Feature | #Instance |
|---|---|---|---|
| Criteo | 39 | 1,327,180 | 45,840,617 |
| Avazu | 23 | 1,544,257 | 40,428,967 |
| ML-1M | 7 | 13,265 | 739,012 |
| ML-Tag | 3 | 90,448 | 2,006,859 |
| Frappe | 10 | 5,392 | 288,609 |

## 4 EXPERIMENT

### 4.1 EXPERIMENTAL SETTING

**Datasets.** We evaluate the proposed RFM on five public datasets, following previous works: Criteo, Avazu, ML-1M, ML-Tag, and Frappe. The statistics of the datasets are shown in Table 2. Due to the page limitation, more details on dataset processing are listed in the Appendix B.

**Metrics.** We adopt AUC (Lobo et al., 2008) and LogLoss (Buja et al., 2005) to evaluate the model performance.

**Baselines.** We compare RFM with the following state-of-the-art models: (1) *First-Order* (**FO**): LR (Richardson et al., 2007); (2) *Second-Order* (**SO**): FwFM (Pan et al., 2018), FmFM (Sun et al., 2021); (3) *High-Order* (**HO**): NFM (He & Chua, 2017), CIN (Lian et al., 2018), CrossNet (Wang et al., 2021), PNN (Qu et al., 2016); (4) *Ensemble* (**EN**): AutoInt+ (Song et al., 2019) (Also known as Transformer), DeepFM Guo et al., xDeepFM (Lian et al., 2018), DCNV2 (Wang et al., 2021); (5) *Adaptive-Order* (**AO**): AFN+ (Cheng et al., 2020), ARM-Net (Cai et al., 2021), EulerNet (Tian et al., 2023). **The description and reproducibility details are presented in Appendices C and D.**

Table 3: Performance comparisons. **Note that a higher AUC or a lower Logloss at the 0.001 level is regarded as significant**, as stated in Tian et al. (2023); Song et al. (2019); Cheng et al. (2016); Guo et al. (2017). "*" denotes that statistical significance for $p < 0.01$ compare to the best baseline. "LL" denotes the LogLoss.

| Type | Model | Criteo | | Avazu | | ML-1M | | ML-Tag | | Frappe | | Effciency | |
|---|---|---|---|---|---|---|---|---|---|---|---|---|---|
| | | AUC | LL | AUC | LL | AUC | LL | AUC | LL | AUC | LL | Params | Latency |
| FO | LR | 0.7900 | 0.4598 | 0.7663 | 0.3879 | 0.8712 | 0.3506 | 0.9303 | 0.3455 | 0.9379 | 0.2858 | 5.39 K | 0.76 ms |
| SO | FwFM | 0.8104 | 0.4414 | 0.7817 | 0.3813 | 0.8934 | 0.3201 | 0.9415 | 0.2761 | 0.9764 | 0.1791 | 91.66 K | 1.02 ms |
| | FmFM | 0.8112 | 0.4408 | 0.7794 | 0.3819 | 0.8942 | 0.3191 | 0.9595 | 0.2255 | 0.9783 | 0.1675 | 93.21 K | 1.21 ms |
| HO | NFM | 0.8066 | 0.4456 | 0.7832 | 0.3784 | 0.8931 | 0.3245 | 0.9578 | 0.2353 | 0.9779 | 0.1722 | 216.14 K | 2.14 ms |
| | CIN | 0.8109 | 0.4424 | 0.7852 | 0.3771 | 0.8913 | 0.3255 | 0.9624 | 0.2125 | 0.9816 | 0.1669 | 362.27 K | 3.79 ms |
| | CrossNet | 0.8123 | 0.4398 | 0.7874 | 0.3767 | 0.8983 | 0.3156 | 0.9647 | 0.2159 | 0.9817 | 0.1611 | 272.31 K | 1.78 ms |
| | PNN | 0.8120 | 0.4399 | 0.7841 | 0.3773 | 0.8953 | 0.3233 | 0.9635 | 0.2197 | 0.9813 | 0.1567 | 113.23 K | 1.58 ms |
| EN | Transformer | 0.8126 | 0.4396 | 0.7841 | 0.3778 | 0.8981 | 0.3195 | 0.9642 | 0.2207 | 0.9810 | 0.1647 | 256.13 K | 3.27 ms |
| | DeepFM | 0.8123 | 0.4399 | 0.7856 | 0.3768 | 0.8973 | 0.3166 | 0.9618 | 0.2264 | 0.9812 | 0.1689 | 252.17 K | 1.61 ms |
| | xDeepFM | 0.8124 | 0.4406 | 0.7874 | 0.3761 | 0.8969 | 0.3187 | 0.9625 | 0.2121 | 0.9819 | 0.1580 | 375.22 K | 5.70 ms |
| | DCNV2 | 0.8129 | 0.4392 | 0.7876 | 0.3757 | 0.8989 | 0.3147 | 0.9649 | 0.2084 | 0.9822 | 0.1531 | 302.99 K | 2.03 ms |
| AO | AFN+ | 0.8125 | 0.4395 | 0.7877 | 0.3756 | 0.8931 | 0.3230 | 0.9607 | 0.2285 | 0.9813 | 0.1697 | 1976.36 K | 3.39 ms |
| | ARM-Net+ | 0.8125 | 0.4395 | 0.7877 | 0.3757 | 0.8969 | 0.3141 | 0.9650 | 0.2096 | 0.9818 | 0.1517 | 1648.16 K | 5.62 ms |
| | EulerNet | 0.8139 | 0.4387 | 0.7879 | 0.3755 | 0.9010 | 0.3098 | 0.9656 | 0.2134 | 0.9832 | 0.1581 | 170.76 K | 1.88 ms |
| | RFM | **0.8147***  | **0.4374***  | **0.7890***  | **0.3749***  | **0.9026***  | **0.3090***  | **0.9667***  | **0.2049***  | **0.9843***  | 0.1506 | 348.17 K | 2.27 ms |

## 4.2 Overall Performance

The overall performance is shown in Table 3. We have the following observations: (1) Low-order models (*i.e.,* LR, FwFM and FmFM) perform worse than high-order models (*i.e.,* NFM, CIN, CrossNet and PNN), due to limited learning capacity. (2) Ensemble methods (*i.e.,* Transformer, DeepFM, xDeepFM, DCNV2) achieve competitive performance across all datasets, showing the effectiveness of integrating MLPs for learning enhanced feature interactions. (3) For adaptive-order models, ARM-Net+ outperforms AFN on the ML-1M, ML-Tag and Frappe datasets, demonstrating the effectiveness of instance-aware interaction learning. Additionally, EulerNet performs very well across all datasets, indicating that the complex vector space is more suitable for learning adaptive-order interactions. (4) RFM consistently outperforms all compared baselines, showing the effectiveness of our proposed self-attentive rotation function for learning *relation-aware* interactions.

For efficiency, we observe that the latency of first-order and second-order models is relatively small due to their simple architectures. The high-order and ensemble models are more time-consuming because they have more complicated architectures. Compared to EulerNet, AFN+ and ARM-Net+ have to incorporate many more parameters to compensate for the limited representation capacity. Note that RFM is sufficiently efficient and is comparable to many efficient approaches (*e.g.,* DCNV2). The complexity of RFM is of the same order as that of the Transformer (See Appendix E).

## 4.3 Further Study

**Ablation Study.** We first analyze how our proposed components influence the performance of RFM. The results are shown in Table 4. We propose four variants as follows: (1) *w/o* AttRo: removing the self-attentive rotation layer, (2) *w/o* AmpNet: removing the modulus amplification network, (3) *w/o* Res: removing the residual in the self-attentive rotation layer, (4) *w/o* Coo Trans: removing the coordinate transformation procedure (See Section 3.2) that directly feeds the angular representations to an MLP. We can see that all these variants underperform the complete RFM, showing that all of our proposed approaches are useful to improve the performance. Specially, the model performance of variant (1) shows a significant decrease, indicating that the self-attentive rotation layer is the core of RFM for learning effective feature interactions. We further present the hyper-parameter studies in Appendix F, and visualize the effect of the modulus amplification network in Appendix I.

Table 4: Components.

| Variant | ML-Tag | | Frappe | |
|---|---|---|---|---|
| | AUC | LogLoss | AUC | LogLoss |
| (0): RFM | 0.9667 | 0.2049 | 0.9843 | 0.1506 |
| (1): *w/o* AttRo | 0.9552 | 0.2454 | 0.9763 | 0.1768 |
| (2): *w/o* AmpNet | 0.9629 | 0.2178 | 0.9804 | 0.1491 |
| (3): *w/o* Res | 0.9635 | 0.2164 | 0.9806 | 0.1620 |
| (4): *w/o* Coo Trans | 0.9637 | 0.2163 | 0.9816 | 0.1611 |

Table 5: Attention and Normalization.

| Variant | ML-Tag | | Frappe | |
|---|---|---|---|---|
| | AUC | LogLoss | AUC | LogLoss |
| (5): *w/o* AttWeight | 0.9656 | 0.2073 | 0.9816 | 0.1533 |
| (6): (1) + *w* DotAtt | 0.9607 | 0.2330 | 0.9813 | 0.1561 |
| (7): *w/o* GN | 0.9626 | 0.2188 | 0.9789 | 0.1678 |
| (8): (7) + *w* LN | 0.9646 | 0.2137 | 0.9820 | 0.1491 |
| (9): (7) + *w* BN | 0.9653 | 0.2089 | 0.9833 | 0.1475 |

Besides, we investigate the effects of our proposed self-attentive rotation function in Table 5. In variant (5), we remove the weight vector (*i.e.*, $w$ in Eq. 5) of rotation-based attention algorithm. Variant (6) replaces the rotation-based attention with the widely used scaled dot-product attention (Vaswani et al., 2017). The performance of both variants shows a notable decrease. This indicates that our proposed rotation-based attention mechanism is more effective for the relation modeling of the angular representations in the complex plane. In variants (7), (8), and (9), we explore the effects of different normalization methods. The results show that GroupNorm is more suitable for learning the feature-wise representations. More ablation study results are presented in Appendix G.

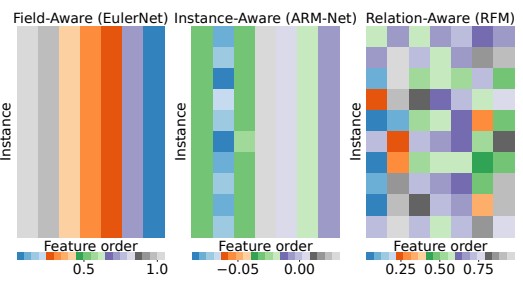 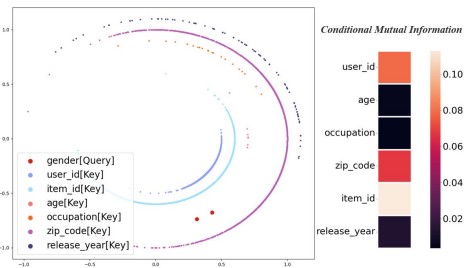

(a) Visualizing the interaction orders.

(b) Visualizing the representations.

Figure 3: Interpretability analysis on the MovieLens-1M dataset.

**Interpretability Analysis.** RFM is capable of adaptively learning the interaction orders from different interaction contexts. Figure 3(a) visualizes the learned orders of different methods. We can observe that the orders in field-aware method are the same for all features within each field (*i.e.*, the columns). The instance-aware method can identity the importance of some features (*i.e.*, $2_{nd}$ column), but cannot capture the dependencies between different fields. In contrast, RFM can learn the varied feature interactions from different contexts. The diversified orders learned from the varying context enable it to capture more effective relationships.

To have an intuitive understanding of our approach, we visualize the representations with a simple case (the embedding dimension $d = 1$) on the MovieLens-1M dataset. As shown in Figure 3(b), the left figure visualizes the query angles (*i.e.*, $\theta_j^Q$ in Eq.5) of the *gender* features and the key angles (*i.e.*, $\theta_j^K$ in Eq.5) of others, and the right figure illustrates the conditional mutual information scores on the *gender* features, representing the strength of each feature field on the ground-truth labels given the *gender* features. We can observe that the fields (*user_id*, *zip_code* and *item_id*) have a strong effect on the results, and they are closely aligned with the *gender* features. For the fields with less importance (*age*, *occupation* and *release_year*), they have no intersecting features with *gender* and the corresponding rotation angles are relatively large. These results indicate that the rotation angles from keys to queries can reflect the importance of feature relationships, which enables RFM to capture the effective feature dependencies for learning varied feature interactions.

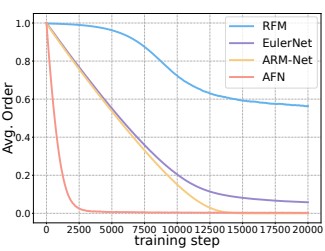 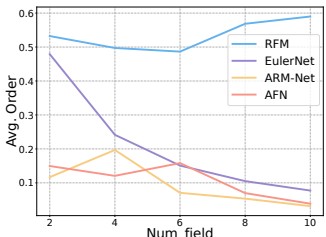 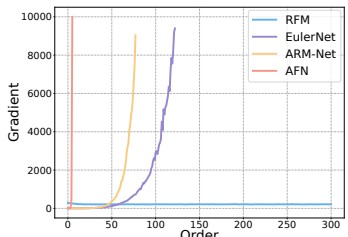

(a) The learning trajectory of average feature order

(b) The average feature order w.r.t the number of feature fields

(c) The gradient w.r.t the maximal interaction order.

Figure 4: Interaction order learning analysis on the Frappe dataset.

**Arbitrary-Order Learning Analysis.** We investigate the arbitrary-order learning capacity of different approaches. Figure 4(a) shows the trajectory of the average feature order (*i.e.,* $\alpha_j$ in Eq. 1) during training. We can see that RFM converges to a relatively large order, while other models tend to approach a zero order. This demonstrates RFM's ability to learn more effective interactions.

Then we probe the learning effectiveness with respect to the number of feature fields (*i.e.,* $m$ in Eq. 1). As shown in Figure 4(b), the average orders learned in EulerNet, AFN and ARM-Net decrease when adding the feature fields. This is due to the fact that the interaction terms exponentially grow with the increasing number of feature fields (*i.e.,* $m$ in Eq. 1). Figure 4(c) shows the gradients with respect to the interaction order (*i.e.,* $\sum_{j=1}^{m} \alpha_j$ in Eq.1). We observe that the gradient in EulerNet, AFN and ARM-Net exponentially grows with the increasing order, leading to the gradient explosion issue when the order reaches a large value. In contrast, the gradient in RFM remains relatively stable, and it is more robust to a large number of feature fields or large interaction orders. These results demonstrate the superiority of our proposed self-attentive rotation function for learning high-order feature interactions. We further provide theoretical analysis in Appendix A.4.

## 5 RELATED WORK

**Feature Interaction Learning.** Learning feature interactions is a fundamental problem in various machine learning tasks, leading to the emergence of several interaction models (Rendle, 2010; Huang et al., 2019; Li et al., 2019; Chen et al., 2019; Yu et al., 2020; Lu et al., 2021). Among them, FM (Rendle, 2010) is the most basic model, using feature embedding vectors to capture second-order interactions. Besides, HOFM (Blondel et al., 2016) introduces a dynamic programming algorithm for higher-order interactions; xDeepFM (Lian et al., 2018) and DCNV2 (Wang et al., 2021) propose intricate interaction architectures to iteratively enumerate the interactions within a predefined order. These methods have significantly improved performance across various applications. However, their reliance on empirically designed orders may hinder accurate learning in real-world contexts. Recent works (Cheng et al., 2020; Cai et al., 2021; Tian et al., 2023) propose to automatically learn the orders from data. However, these methods cannot capture the feature dependencies in varying contexts, which diminishes the model's capacity. Further, they suffer from the exponential explosion issue, making them unsuitable for scenarios with numerous features or high orders. Different from them, we utilize the attentive rotations to model complicated interactions, which can adaptively capture the feature dependencies and surpass the scale limits of the interaction order in existing studies.

**Representation Learning with Complex Vectors.** In the literature, numerous approaches are proposed to learn the relations in the complex vector space for enhancing the representations. Especially, WaveMLP (Tang et al., 2022) represents each image patch as a wave to capture the dynamic vision semantics. Additionally, RotatE (Sun et al., 2019) defines each relation of a knowledge graph as a rotation from the source entity to the target entity. RoPE (Su et al., 2021) and XPOS (Sun et al., 2022) leverage a two-dimensional pairwise rotation method to improve the position embedding of Transformers. In the area of feature interaction learning, EulerNet (Tian et al., 2023) proposes utilizing Euler's formula to adaptively learn the arbitrary-order feature interactions. These approaches provide a new way to enhance representation learning in a variety of machine learning tasks.

## 6 CONCLUSION

In this paper, we propose a novel Rotative Factorization Machine (**RFM**) for better modeling feature interactions in the prediction tasks. Unlike prior work, RFM represents each feature as a polar angle in the complex plane and converts the interactions into the complex rotations, avoiding the exponential explosion of the interaction terms. In RFM, the rotation coefficients are modeled through a rotation-based attention mechanism, which can adaptively learn the interaction orders from different interaction contexts. Moreover, we propose a modulus amplification network to learn the modulus of the complex features for further enhancing the feature interaction learning. As the main contribution, we propose a novel self-attentive rotation function to model complicated feature interactions, providing a way to learn the unbounded interaction orders adaptively from the corresponding interaction contexts. As future work, we consider extending the RFM to handle sequential, spatial, and other forms of structured data, and deploy it across multiple domains and tasks.

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

# A THEORETICAL ANALYSIS

## A.1 TENSOR-FORM ATTENTION CALCULATION

In this section, we prove that the tensor-form calculation of Eq. 5 equivalent to Eq. 9. Note that the element in the $j$-th row and $l$-th column can be given as:

$$
\begin{aligned}
A_{j,l} &= \text{Sigmoid}\left(\text{Re}\left[\left(\exp(i\boldsymbol{Q})\text{diag}(\boldsymbol{w})\right)\exp(-i\boldsymbol{K})^{\top}\right]_{j,l}\right) \\
&= \text{Sigmoid}\left(\text{Re}\left[\left(\exp(i\boldsymbol{Q})\text{diag}(\boldsymbol{w})\right)\exp(-i\boldsymbol{K})^{\top}\right]_{j,l}\right) \\
&= \text{Sigmoid}\left(\text{Re}\left[\left(\exp(i\boldsymbol{Q})\text{diag}(\boldsymbol{w})\right)_{j}\left(\exp(-i\boldsymbol{K})_{l}\right)^{\top}\right]\right) \\
&= \text{Sigmoid}\left(\text{Re}\left[\left(\exp(i\boldsymbol{Q}_{j})\odot \boldsymbol{w}^{\top}\right)\exp(-i\boldsymbol{K}_{l})^{\top}\right]\right) \\
&= \text{Sigmoid}\left(\text{Re}\left[\boldsymbol{w}^{\top}\left(\exp(i\boldsymbol{Q}_{j})\odot\exp(-i\boldsymbol{K}_{l})\right)^{\top}\right]\right) \\
&= \text{Sigmoid}\left(\text{Re}\left[\boldsymbol{w}^{\top}\cos(\boldsymbol{Q}_{j}^{\top}-\boldsymbol{K}_{l}^{\top})+i\boldsymbol{w}^{\top}\sin(\boldsymbol{Q}_{j}^{\top}-\boldsymbol{K}_{l}^{\top})\right]\right) \\
&= \text{Sigmoid}\left(\boldsymbol{w}^{\top}\cos(\boldsymbol{\theta}_{j}^{Q}-\boldsymbol{\theta}_{l}^{K})\right) = \alpha_{j,l}.
\end{aligned}
$$

Therefore, the matrix $\boldsymbol{A}$ calculates all pairwise attention scores of $\{\alpha_{j,l}|\forall j,l \in \{1,...,m\}\}$.

## A.2 PROOF OF THEORME 3.1

We first investigate the properties in the high-dimensional vector space:

**Lemma A.1.** *If $d$-dimensional embeddings $\{\boldsymbol{\theta}_{j}\}_{j=1}^{m}$ are L2-regularized such that $||\boldsymbol{\theta}_{j}||_{2} \leq 1, \forall j \in \{1,...,m\}$, let $|\theta_{m}| = \max_{j,l}|\theta_{j,l}|$ denote the max absolute element of the embeddings. Then we have $\Pr(|\theta_{m}| \leq (\sqrt{(4\ln d)/(d-1)}) \geq 1 - \mathcal{O}(m/d)$.*

*Proof.* Since $||\boldsymbol{\theta}_{j}||_{2} \leq 1, \forall j \in [1,m]$, it indicates that all embeddings are bounded in a $d$-dimensional unit ball. Let $V(d)$ denote the volume of the $d$-dimensional unit ball. We first calculate $\Pr(|\theta| > \mathcal{O}(\sqrt{(4\ln d)/(d-1)})$. The upper bound of this volume can be given as:

$$
\begin{aligned}
\mathcal{V} &= \int_{\sqrt{(4\ln d)/(d-1)}}^{1} (1-x^2)^{\frac{d-1}{2}} V(d-1) dx \\
&\leq \int_{\sqrt{(4\ln d)/(d-1)}}^{1} \frac{x\sqrt{d-1}}{\sqrt{4\ln d}}(1-x^2)^{\frac{d-1}{2}} V(d-1) dx \\
&\leq \frac{V(d-1)}{\sqrt{4(d-1)\ln d}} e^{-\frac{4\ln d}{2}} \\
&= \frac{V(d-1)}{2d^2\sqrt{(d-1)\ln d}}.
\end{aligned}
$$

Obviously, the cylinder with a height of $1$ and radius of $\sqrt{1-\frac{1}{d-1}}$ is bounded within a $d$-dimensional hemisphere (volume of $\mathcal{D}$), and thus we have:

$$
\mathcal{D} \geq V(d-1)(1-\frac{1}{d-1})^{\frac{d-1}{2}}\frac{1}{\sqrt{d-1}} \geq \frac{V(d-1)}{2\sqrt{d-1}}.
$$

Therefore, we have:

$$\Pr(|\theta| > (\sqrt{(4\ln d)/(d-1)})) = \frac{\mathcal{V}}{\mathcal{D}} \leq \frac{1}{d^2\sqrt{\ln d}} < \mathcal{O}(\frac{1}{d^2}),$$

$$\Pr(|\theta_m| \leq (\sqrt{(4\ln d)/(d-1)})) = 1 - \Pr\Big(\bigcup_{j,l}|\theta|_{j,l} > (\sqrt{(4\ln d)/(d-1)})\Big)$$

$$\geq 1 - \sum_{j,l}\Pr(|\theta|_{j,l} > (\sqrt{(4\ln d)/(d-1)}))$$

$$= 1 - md\cdot\mathcal{O}(\frac{1}{d^2}) = \mathcal{O}(1 - \frac{m}{d}).$$

$\square$

**Lemma A.2.** *For any given order vector $\boldsymbol{\alpha} \in \{0,1\}^m$ and any input features $\{\boldsymbol{\theta}_j\}_{j=1}^m \in \mathbb{R}^{m\times d}$, the rotation-based interaction pattern $\boldsymbol{\Delta}_G = \mathcal{H}(e^{i\alpha_1\boldsymbol{\theta}_1} \odot e^{i\alpha_2\boldsymbol{\theta}_2} \odot \cdots \odot e^{i\alpha_m\boldsymbol{\theta}_m})$ can be degenerated to the inner product based interaction $\boldsymbol{\Delta}_F = \boldsymbol{e}_1^{\alpha_1} \odot \boldsymbol{e}_2^{\alpha_2} \odot \cdots \boldsymbol{e}_m^{\alpha_m}$ with a max error of $\mathcal{R} = max(|\boldsymbol{\Delta}_G - \boldsymbol{\Delta}_F|) \leq \mathcal{O}(\sum_{j=1}^m \alpha_j|\theta_m|)$, where $\mathcal{H}(\boldsymbol{z}) = \mathrm{Re}(\boldsymbol{z}) + \mathrm{Im}(\boldsymbol{z})$.*

*Proof.* Note that $|\mathcal{H}(\boldsymbol{z})| \leq ||\mathrm{Re}(\boldsymbol{z})| + |\mathrm{Im}(\boldsymbol{z})||$ and $\cos(\alpha_j\boldsymbol{\theta}_j) = \cos^{\alpha_j}(\boldsymbol{\theta}_j)$ if $\alpha_j \in \{0,1\}$. Let $\boldsymbol{e}_j := \cos(\boldsymbol{\theta}_j) \in \mathbb{R}^d$, and we have:

$$|\boldsymbol{\Delta}_G - \boldsymbol{\Delta}_F| = |\mathcal{H}(e^{i\alpha_1\boldsymbol{\theta}_2} \odot e^{i\alpha_2\boldsymbol{\theta}_2} \odot \cdots \odot e^{i\alpha_m\boldsymbol{\theta}_m}) - \boldsymbol{e}_1^{\alpha_1} \odot \boldsymbol{e}_2^{\alpha_2} \odot \cdots \boldsymbol{e}_m^{\alpha_m}|$$

$$= \left|\mathcal{H}\left(\prod_{j=1}^m \Big(\cos(\alpha_j\boldsymbol{\theta}_j) + i\sin(\alpha_j\boldsymbol{\theta}_j)\Big)\right) - \prod_{j=1}^m \cos^{\alpha_j}(\boldsymbol{\theta}_j)\right|$$

$$= \left|\mathcal{H}\left(\prod_{j=1}^m \Big(\cos(\alpha_j\boldsymbol{\theta}_j) + i\sin(\alpha_j\boldsymbol{\theta}_j)\Big)\right) - \prod_{j=1}^m \cos(\alpha_j\boldsymbol{\theta}_j)\right|$$

$$= \left|\mathcal{H}\left(\sum_{l=1}^m \sum_{p\in C_m^l}\prod_{t=1}^m \Big(i^{p_t}\cos^{1-p_t}(\alpha_t\boldsymbol{\theta}_t)\sin^{p_t}(\alpha_t\boldsymbol{\theta}_t)\Big)\right) + \prod_{j=1}^m \cos(\alpha_j\boldsymbol{\theta}_j) - \prod_{j=1}^m \cos(\alpha_j\boldsymbol{\theta}_j)\right|$$

$$= \left|\mathcal{H}\left(\sum_{l=1}^m \sum_{p\in C_m^l}\prod_{t=1}^m \Big(i^{p_t}\cos^{1-p_t}(\alpha_t\boldsymbol{\theta}_t)\sin^{p_t}(\alpha_t\boldsymbol{\theta}_t)\Big)\right) + \mathbf{1} - \mathbf{1}\right|$$

$$\leq \left|\left(\sum_{l=1}^m \sum_{p\in C_m^l}\prod_{t=1}^m \Big(|\cos^{1-p_t}(\alpha_t\boldsymbol{\theta}_t)| \odot |\sin^{p_t}(\alpha_t\boldsymbol{\theta}_t)|\Big)\right) + \mathbf{1} - \mathbf{1}\right|$$

$$\leq \left|\left(\sum_{l=1}^m \sum_{p\in C_m^l}\prod_{t=1}^m \Big(\mathbf{1} \odot |\sin^{p_t}(\alpha_t\boldsymbol{\theta}_t)|\Big)\right) + \mathbf{1} - \mathbf{1}\right|$$

$$= \left|\prod_{j=1}^m \Big(\mathbf{1} + |\sin(\alpha_j\boldsymbol{\theta}_j)|\Big) - \mathbf{1}\right|$$

Here $C_m^l$ denotes the set of indices representing the combinations that select $l$ elements from a set of size $m$, *e.g.*, $C_3^2 = \{[0,1,1],[1,0,1],[1,1,0]\}$. Therefore, we have:

$$\mathcal{R} \leq max\left(\left|\prod_{j=1}^m \Big(\mathbf{1} + |\sin(\alpha_j\boldsymbol{\theta}_j)|\Big) - \mathbf{1}\right|\right) \leq \left|\Big(1 + |\theta_m|\Big)^{\sum_{j=1}^m \alpha_j} - \mathbf{1}\right| = \mathcal{O}(\sum_{j=1}^m \alpha_j|\theta_m|).$$

$\square$

As discussed in the Section 2, for the $j$-th feature field, each feature is represented as a one-hot vector $\boldsymbol{x}_j \in \{0,1\}^{n_j}$, where $n_j$ is the feature number in the $j$-th field, and $N = \sum_{j=1}^m n_j$ is the

total number of features. Afterwards, the embedding look-up operation $E(\cdot)$ is employed to map the one-hot vector $\boldsymbol{x}_j$ to a low-dimensional embedding $\boldsymbol{e}_j$, *i.e.*, $\boldsymbol{e}_j = E(\boldsymbol{x}_j)$. Formally, the one-hot encoded vector $\boldsymbol{x}_j$ of the $l$-th feature in the $j$-th field is defined as $\boldsymbol{x}_j[k] = \begin{cases} 1, & k = l \\ 0, & k \neq l \end{cases}$, and we define the ***index*** of the $l$-th feature in the $j$-th field as $\mathrm{ID}(\boldsymbol{x}_j) = \sum_{k=1}^{j-1} n_k + l$, its inverse function as $\mathrm{OneHot}(\sum_{k=1}^{j-1} n_k + l) = \boldsymbol{x}_j$, and the function $\mathcal{G}(s) = \arg\min_j \sum_{k=1}^{j} n_k \geq s$ returns the field index of the global index $s$. We place all features along an axis, and the truth table $\mathcal{T}$ of the feature dependency function $f$ is denoted as a matrix $\mathcal{T} \in \{0,1\}^{N \times N}$, where:

$$\mathcal{T}(s,t) = \begin{cases} f\Big(E\big(\mathrm{OneHot}(s)\big), E\big(\mathrm{OneHot}(t)\big)\Big), & \mathcal{G}(s) \neq \mathcal{G}(t) \\ 0, & \mathcal{G}(s) = \mathcal{G}(t) \end{cases}$$

Given the input feature embeddings $\{\boldsymbol{\theta}_j\}_{j=1}^m$, and their one-hot representations $\{\boldsymbol{x}_j\}_{j=1}^m$, we can obtain the relation vector $\boldsymbol{r}_j$ of the feature $\boldsymbol{x}_j$:

$$\boldsymbol{r}_j[k] = \begin{cases} \mathcal{T}\Big(k, \mathrm{ID}(\boldsymbol{x}_j)\Big), & \mathcal{G}(k) \neq j \\ \frac{1}{2} + \frac{1}{2} \cdot \boldsymbol{x}_j[k - \sum_{l=1}^{j-1} n_k], & \mathcal{G}(k) = j \end{cases}$$

where $k \in \{1, ..., N\}$. The vector $\boldsymbol{r}_j$ measures the relation between the feature $\boldsymbol{e}_j$ and the features from other fields. Meanwhile, the feature dimensions of the same field are naturally masked with $\frac{1}{2}$, except for itself, which has a value of $1$. We use the vector $\boldsymbol{r}_j$ as the auxiliary dimensions for the input features, $\tilde{\boldsymbol{\theta}}_j = [\boldsymbol{\theta}_j, \epsilon \cdot \boldsymbol{r}_j]$, where $\epsilon$ is a sufficiently small number. We construct the matrix $\boldsymbol{M}$ as follows:

$$\boldsymbol{M} = \left[ \begin{array}{cc} \mathbb{O}_{N \times d} & \frac{\pi}{\epsilon} \cdot \boldsymbol{I}_{N \times N} \end{array} \right] \in \mathbb{R}^{N \times (d+N)}.$$

Here $\mathbb{O}$ is an all-zero matrix. We have $\boldsymbol{M}\tilde{\boldsymbol{\theta}}_j = \pi \cdot \boldsymbol{r}_j$. Here we set all query matrices and key matrices as $\boldsymbol{W}_j^Q = \boldsymbol{W}_j^K = \boldsymbol{M}, \forall j = \{1, 2, ..., m\}$, and set all the value matrices $\boldsymbol{W}_j^V$ as the identity matrix $\boldsymbol{I}$. We construct $m$ attention heads, each measuring the relationship between the features in the $j$-th field ($j \in [1, m]$) and all the other features. Formally, the projection matrices $\boldsymbol{H}_j^Q, \boldsymbol{H}_j^K, \boldsymbol{H}_j^V$ are defined as:

$$\boldsymbol{H}_j^Q = \boldsymbol{H}_j^K = \left[ \begin{array}{ccccc} \mathbb{O}_{n_j \times n_1} & \cdots & \boldsymbol{I}_{n_j \times n_j} & \cdots & \mathbb{O}_{n_j \times n_m} \end{array} \right]^\top \in \mathbb{R}^{N \times n_j},$$

$$\boldsymbol{H}_j^V = \left[ \begin{array}{ccccc} \boldsymbol{I}_{d \times d} & \mathbb{O}_{d \times n_1} & \cdots & \mathbb{O}_{d \times n_j} & \cdots & \mathbb{O}_{d \times n_m} \end{array} \right]^\top \in \mathbb{R}^{(d+N) \times d}$$

In this way, the values are projected to the original features $\boldsymbol{V}^{(j)} = \{\boldsymbol{\theta}_k\}_{k=1}^m$. The queries and keys of the $j$-th head are projected to the following: $\boldsymbol{Q}^{(j)} = \boldsymbol{K}^{(j)} = \{\pi \cdot \boldsymbol{r}_k^{(j)}\}_{k=1}^m$. Assume that the one-hot vector $\boldsymbol{x}_j = [0, 0, \cdots, \underbrace{1}_{l-th \text{ element}}, 0, 0, \cdots]^\top$, the projected vector $\boldsymbol{r}_k^{(j)}$ takes the following form:

$$\boldsymbol{r}_k^{(j)} = \begin{cases} [\frac{1}{2}, \frac{1}{2}, \cdots, \underbrace{1}_{l-th \text{ element}}, \frac{1}{2}, \frac{1}{2}, \cdots]^\top, & k = j \\ [0, 1, \cdots, \underbrace{1}_{\mathcal{T}\big(\mathrm{ID}(\boldsymbol{x}_j), \mathrm{ID}(\boldsymbol{x}_k)\big)}, 0, 1, \cdots]^\top, & k \neq j \end{cases}$$

We set the weight vector $\boldsymbol{w} = [S, ..., S]^\top$, and $S > 0$ is a sufficiently large number. Note that $\cos(\pm\frac{\pi}{2}) = 0$. Considering the attention score from $j$-th query in $j$-th attention head, we have:

$$\begin{aligned} \alpha_{j,l}^{RFM} &= \mathrm{Sigmoid}\Big(\boldsymbol{w}^\top \cos(\pi \cdot \boldsymbol{r}_j^{(j)} - \pi \cdot \boldsymbol{r}_l^{(j)})\Big) \\ &= \mathrm{Sigmoid}\Big(S \cdot \cos\Big(\pi - \pi \cdot \mathcal{T}\big(\mathrm{ID}(\boldsymbol{x}_j), \mathrm{ID}(\boldsymbol{x}_l)\big)\Big)\Big) \\ &= \mathcal{T}\Big(\mathrm{ID}(\boldsymbol{x}_j), \mathrm{ID}(\boldsymbol{x}_l)\Big) = f(\boldsymbol{e}_j, \boldsymbol{e}_l). \end{aligned}$$

According to Eq. 7, we have:

$$\hat{\boldsymbol{\theta}}_j = \sum_{l=1}^{m} \alpha_{j,l}^{RFM} \boldsymbol{\theta}_l^V = \sum_{l=1}^{m} f(\boldsymbol{e}_j, \boldsymbol{e}_l)\boldsymbol{\theta}_l.$$

In this scheme, we only consider the $j$-th query in the $j$-th attention head ($j \in [1, m]$), and set the weight $\boldsymbol{u}$ (See in Eq. 19) as the identity matrix $\boldsymbol{I}$. According to Eq. 17, omitting the activation function yields the following expression for the output of RFM:

$$\begin{aligned}
\hat{y} &= \boldsymbol{u}^\top (\cos(\hat{\boldsymbol{\theta}}_j) + \sin(\hat{\boldsymbol{\theta}}_j)) \\
&= \cos(\sum_{l=1}^{m} f(\boldsymbol{e}_j, \boldsymbol{e}_l)\boldsymbol{\theta}_l) + \sin(\sum_{l=1}^{m} f(\boldsymbol{e}_j, \boldsymbol{e}_l)\boldsymbol{\theta}_l) \\
&= \mathcal{H}\Big( \cos(\sum_{l=1}^{m} f(\boldsymbol{e}_j, \boldsymbol{e}_l)\boldsymbol{\theta}_l) + i \sin(\sum_{l=1}^{m} f(\boldsymbol{e}_j, \boldsymbol{e}_l)\boldsymbol{\theta}_l)\Big) \\
&= \mathcal{H}\Big( \exp(i \sum_{l=1}^{m} f(\boldsymbol{e}_j, \boldsymbol{e}_l)\boldsymbol{\theta}_l)\Big) \\
&= \mathcal{H}(e^{if(\boldsymbol{e}_j, \boldsymbol{e}_1)\boldsymbol{\theta}_1} \odot e^{if(\boldsymbol{e}_j, \boldsymbol{e}_2)\boldsymbol{\theta}_2} \odot \cdots \odot e^{if(\boldsymbol{e}_j, \boldsymbol{e}_m)\boldsymbol{\theta}_m}).
\end{aligned}$$

Since the construction of the order is independent of the input features $\{\boldsymbol{e}_j = \boldsymbol{\theta}_j\}_{j=1}^{m}$, the theorem 3.1 is proved by combining lemma A.1 and lemma A.2.

### A.3 FIELD-AWARE AND INSTANCE-AWARE INTERACTION LEARNING

The proof is equivalent to proving the following two lemmas:

**Lemma A.3.** *If embeddings $\{\boldsymbol{\theta}_j\}_{j=1}^{m}$ are L2-regularized such that $\|\theta_j\|_2 \leq 1, \forall j \in \{1, ..., m\}$, then for any given order $\boldsymbol{\alpha} \in \{0, 1\}^m$, RFM can model the interaction pattern $\boldsymbol{\Delta}_F = \boldsymbol{e}_1^{\alpha_1} \odot \boldsymbol{e}_2^{\alpha_2} \odot \cdots \boldsymbol{e}_m^{\alpha_m}$.*

*Proof.* We add an auxiliary dimension to the input features, $\tilde{\boldsymbol{\theta}}_j = [\boldsymbol{\theta}_j, \epsilon]$, where $\epsilon$ is a sufficiently small number. We construct two types of matrices: $\boldsymbol{N} = \mathbb{O}_{(d+1) \times (d+1)}$ is an all-zero matrix with a shape of $(d+1) \times (d+1)$, and $\boldsymbol{M}$ is defined by the following:

$$\boldsymbol{M} = \begin{bmatrix} 0 & \cdots & 0 \\ \vdots & \ddots & \vdots \\ 0 & \cdots & \frac{\pi}{\epsilon} \end{bmatrix} \in \mathbb{R}^{(d+1) \times (d+1)}.$$

In this way, we have $\boldsymbol{M}\tilde{\boldsymbol{\theta}}_j = [0, ..., \pi]^\top$ and $\boldsymbol{N}\tilde{\boldsymbol{\theta}}_j = [0, ..., 0]^\top$. Here, we set all query matrices as $\boldsymbol{W}_j^Q = \boldsymbol{N}, \forall j = \{1, 2, ..., m\}$. Given the order vector $\alpha$, the key matrices are set by the following rule:

$$\boldsymbol{W}_j^K = \begin{cases} \boldsymbol{M}, & \alpha_j = 0 \\ \boldsymbol{N}, & \alpha_j = 1 \end{cases}$$

In this way, the matrices of the queries are mapped to a zero space, *i.e.*, $\boldsymbol{\theta}_j^Q = \boldsymbol{W}_j^Q \tilde{\boldsymbol{\theta}}_j = \boldsymbol{N}\tilde{\boldsymbol{\theta}} = \boldsymbol{0}$. As for the keys, when $j$ satisfies $\alpha_j = 0$, the transformed vector $\boldsymbol{\theta}_j^K = \boldsymbol{W}_j^K \tilde{\boldsymbol{\theta}}_j = \boldsymbol{M}\tilde{\boldsymbol{\theta}}_j = [0, ..., \pi]^\top$; when $j$ satisfies $\alpha_j = 1$, $\boldsymbol{\theta}_j^K = \boldsymbol{W}_j^K \tilde{\boldsymbol{\theta}}_j = \boldsymbol{N}\tilde{\boldsymbol{\theta}}_j = \boldsymbol{0}$. We set the weight vector $\boldsymbol{w} = S \cdot [0, ..., 1]^\top$, and $S > 0$ is a sufficiently large number. Consider the attention score from $j$-th query, we have:

$$\alpha_{j,l}^{RFM} = \mathrm{RotAtt}(\boldsymbol{Q}_j, \boldsymbol{K}_l) = \mathrm{Sigmoid}\Big(\boldsymbol{w}^\top \cos(\boldsymbol{\theta}_j^Q - \boldsymbol{\theta}_l^K)\Big) = \begin{cases} \mathrm{Sigmoid}(-S) = 0, & \alpha_l = 0 \\ \mathrm{Sigmoid}(S) = 1, & \alpha_l = 1 \end{cases}$$

Therefore, we have $\boldsymbol{\alpha}_j^{RFM} = \boldsymbol{\alpha}$. Furthermore, we define the value matrix as follows:

$$\boldsymbol{W}_j^V = \begin{bmatrix} 1 & \cdots & 0 & 0 \\ \vdots & \ddots & \vdots & \vdots \\ 0 & \cdots & 1 & 0 \end{bmatrix} \in \mathbb{R}^{d \times (d+1)}.$$

Therefore $\boldsymbol{\theta}_j^V = \boldsymbol{W}_j^V \tilde{\boldsymbol{\theta}}_j = \boldsymbol{\theta}_j$. According to Eq. 7, we have:

$$\hat{\boldsymbol{\theta}}_j = \sum_{l=1}^m \alpha_{j,l}^{RFM} \boldsymbol{\theta}_l^V = \sum_{l=1}^m \alpha_l \boldsymbol{\theta}_l.$$

In this scheme, all $\hat{\boldsymbol{\theta}}_j$ are the same, and we only consider a single output of rotated angles. We remove the amplification network, and set the weight $\boldsymbol{u}$ (See Eq. 19) as the identity matrix $\boldsymbol{I}$. According to Eq. 17, when omitting the activation function, the output of RFM can be given as:

$$\begin{aligned}
\hat{y} &= \boldsymbol{u}^\top (\cos(\hat{\boldsymbol{\theta}}_l) + \sin(\hat{\boldsymbol{\theta}}_l)) \\
&= \cos(\sum_{l=1}^m \alpha_l \boldsymbol{\theta}_l) + \sin(\sum_{l=1}^m \alpha_l \boldsymbol{\theta}_l) \\
&= \mathcal{H}\Big( \cos(\sum_{l=1}^m \alpha_l \boldsymbol{\theta}_l) + i \sin(\sum_{l=1}^m \alpha_l \boldsymbol{\theta}_l) \Big) \\
&= \mathcal{H}\Big( \exp(i \sum_{l=1}^m \alpha_l \boldsymbol{\theta}_l) \Big) \\
&= \mathcal{H}(e^{i\alpha_1 \boldsymbol{\theta}_1} \odot e^{i\alpha_2 \boldsymbol{\theta}_2} \odot \cdots \odot e^{i\alpha_m \boldsymbol{\theta}_m}).
\end{aligned}$$

Therefore, Lemma A.3 is proved by combining Lemma A.1 and Lemma A.2. $\qquad\square$

**Lemma A.4.** *If embeddings $\{\boldsymbol{\theta}_j\}_{j=1}^m$ are L2-regularized such that $||\boldsymbol{\theta}_j||_2 \leq 1, \forall j \in \{1, ..., m\}$, RFM can model the interaction pattern $\boldsymbol{\Delta}_I = \boldsymbol{e}_1^{\alpha_1} \odot \boldsymbol{e}_2^{\alpha_2} \odot \cdots \boldsymbol{e}_m^{\alpha_m}$, where $\alpha_j = f(\boldsymbol{e}_j)$ and $f : \mathbb{R}^d \to \{0, 1\}$ is an instance importance function.*

*Proof.* Given the set of input feature embeddings $\{\boldsymbol{e}_j = \boldsymbol{\theta}_j\}_{j=1}^m$, we add an auxiliary dimension to the input features, $\tilde{\boldsymbol{\theta}}_j = [\boldsymbol{\theta}_j, \epsilon \cdot f(\boldsymbol{e}_j)]$, where $\epsilon$ is a sufficiently small number. We construct two types of matrices: $\boldsymbol{N} = \mathbb{O}_{(d+1)\times(d+1)}$ is an all-zeros matrix with the shape of $(d+1) \times (d+1)$, and $\boldsymbol{M}$ is defined by the following:

$$\boldsymbol{M} = \begin{bmatrix} 0 & \cdots & 0 \\ \vdots & \ddots & \vdots \\ 0 & \cdots & \frac{\pi}{\epsilon} \end{bmatrix} \in \mathbb{R}^{(d+1)\times(d+1)}.$$

We have $\boldsymbol{M}\tilde{\boldsymbol{\theta}}_j = [0, ..., \pi \cdot f(\boldsymbol{e}_j)]^\top$ and $\boldsymbol{N}\tilde{\boldsymbol{\theta}}_j = [0, ..., 0]^\top$. Here we set all query matrices as $\boldsymbol{W}_j^Q = \boldsymbol{N}, \forall j = \{1, 2, ..., m\}$ and key matrices as $\boldsymbol{W}_j^K = \boldsymbol{M}, \forall j = \{1, 2, ..., m\}$. We set the weight vector $\boldsymbol{w} = [0, ..., -S]^\top$, and $S > 0$ is a sufficiently large number. In this way, the matrices for the queries are mapped into a zero space, *i.e.*, $\boldsymbol{\theta}_j^Q = \boldsymbol{W}_j^Q \tilde{\boldsymbol{\theta}}_j = \boldsymbol{N}\tilde{\boldsymbol{\theta}} = \boldsymbol{0}$, and the keys are $\boldsymbol{\theta}_j^K = \boldsymbol{W}_j^K \tilde{\boldsymbol{\theta}}_j = \boldsymbol{M}\tilde{\boldsymbol{\theta}}_j = [0, ..., \pi \cdot f(\boldsymbol{e}_j)]^\top$. Since $f(\boldsymbol{e}_j) \in \{0, 1\}$, thus we have $\text{Sigmoid}\Big( - S \cdot \cos \Big( \pi \cdot f(\boldsymbol{e}_j) \Big) \Big) = f(\boldsymbol{e}_j)$. Consider the attention score from $j$-th query, we have:

$$\alpha_{j,l}^{RFM} = \text{Sigmoid}\Big( \boldsymbol{w}^\top \cos(\boldsymbol{\theta}_j^Q - \boldsymbol{\theta}_l^K) \Big) = \text{Sigmoid}\Big( - S \cdot \cos \Big( \pi \cdot f(\boldsymbol{e}_l) \Big) \Big) = f(\boldsymbol{e}_l).$$

Further, we set the value matrices as the following:

$$\boldsymbol{W}_j^V = \begin{bmatrix} 1 & \cdots & 0 & 0 \\ \vdots & \ddots & \vdots & \vdots \\ 0 & \cdots & 1 & 0 \end{bmatrix} \in \mathbb{R}^{d\times(d+1)}.$$

Therefore $\boldsymbol{\theta}_j^V = \boldsymbol{W}_j^V \tilde{\boldsymbol{\theta}}_j = \boldsymbol{\theta}_j$. According to Eq. 7, we have:

$$\hat{\boldsymbol{\theta}}_j = \sum_{l=1}^m \alpha_{j,l}^{RFM} \boldsymbol{\theta}_j^V = \sum_{l=1}^m \alpha_l \boldsymbol{\theta}_l.$$

In this scheme, all $\hat{\boldsymbol{\theta}}_j$ are the same; we only consider a single output of rotated angles and set the weight $\boldsymbol{u}$ (See Eq. 19) as the identity matrix $\boldsymbol{I}$. According to Eq. 17, when the activation function is omitted, the output of RFM can be given as:

$$
\begin{aligned}
\hat{y} &= \boldsymbol{u}^\top (\cos(\hat{\boldsymbol{\theta}}_l) + \sin(\hat{\boldsymbol{\theta}}_l)) \\
&= \cos(\sum_{l=1}^m \alpha_l \boldsymbol{\theta}_l) + \sin(\sum_{l=1}^m \alpha_l \boldsymbol{\theta}_l) \\
&= \mathcal{H}\Big( \cos(\sum_{l=1}^m \alpha_l \boldsymbol{\theta}_l) + i \sin(\sum_{l=1}^m \alpha_l \boldsymbol{\theta}_l) \Big) \\
&= \mathcal{H}\Big( \exp(i \sum_{l=1}^m \alpha_l \boldsymbol{\theta}_l) \Big) \\
&= \mathcal{H}(e^{i\alpha_1 \boldsymbol{\theta}_1} \odot e^{i\alpha_2 \boldsymbol{\theta}_2} \odot \cdots \odot e^{i\alpha_m \boldsymbol{\theta}_m}) \\
&= \mathcal{H}(e^{if(\boldsymbol{e}_1)\boldsymbol{\theta}_1} \odot e^{if(\boldsymbol{e}_2)\boldsymbol{\theta}_2} \odot \cdots \odot e^{if(\boldsymbol{e}_m)\boldsymbol{\theta}_m}).
\end{aligned}
$$

Since the construction of the order is independent of the input features $\{\boldsymbol{e}_j = \boldsymbol{\theta}_j\}_{j=1}^m$, lemma A.4 is proven by combining lemma A.1 and lemma A.2. $\qquad\square$

### A.4 GRADIENT ANALYSIS

In this section, we analyze and compare the gradient properties of RFM with traditional feature interaction methods. Specifically, we examine the gradient with respect to the number of feature fields (*i.e.,* denoted as $m$ in Eq. 1). In the subsequent theoretical analysis, we will prove that our method exhibits, at most, linear growth in the gradient with respect to the field number. Conversely, in traditional feature interaction approaches, the gradient exhibits exponential growth with respect to the field number . For ease of analysis, we formulate the learning function of our approach as:

$$
\begin{aligned}
\boldsymbol{G} &= \text{Attention}(\boldsymbol{Q}, \boldsymbol{K}, \boldsymbol{V}) \\
&= \sigma \left( \text{Re} \left[ (\exp(i\boldsymbol{Q}) \,\text{diag}(\boldsymbol{\omega})) \exp(-i\boldsymbol{K})^\top \right] \right) \cdot \boldsymbol{V} \\
y &= f(\cos(\boldsymbol{G})) + f(\sin(\boldsymbol{G}))
\end{aligned}
$$

For ease of mathematical illustration, we use the notation $\boldsymbol{X}_j$ to denote the original input feature embedding $\boldsymbol{e}_j$. We first calculate the gradient of our approach, *i.e.,* $\frac{\partial y}{\partial X}$.

Let $\boldsymbol{X} = \begin{bmatrix} \boldsymbol{X_1} & & & \\ & \boldsymbol{X_2} & & \\ & & \ddots & \\ & & & \boldsymbol{X_m} \end{bmatrix} \in \mathbb{R}^{\text{dm} \times \text{m}}, \qquad \begin{aligned} [\boldsymbol{W_1^Q}, \cdots, \boldsymbol{W_m^Q}] &= \tilde{\boldsymbol{B}} \in \mathbb{R}^{\text{d} \times \text{dm}} \\ [\boldsymbol{W_1^K}, \cdots, \boldsymbol{W_m^K}] &= \tilde{\boldsymbol{C}} \in \mathbb{R}^{\text{d} \times \text{dm}} \\ [\boldsymbol{W_1^V}, \cdots, \boldsymbol{W_m^V}] &= \tilde{\boldsymbol{D}} \in \mathbb{R}^{\text{d} \times \text{dm}} \end{aligned}$.

So $\boldsymbol{Q} = (\tilde{\boldsymbol{B}}\boldsymbol{X})^\top \quad \boldsymbol{K} = (\tilde{\boldsymbol{C}}\boldsymbol{X})^\top \quad \boldsymbol{V} = (\tilde{\boldsymbol{D}}\boldsymbol{X})^\top$.

Remark $\text{Re}\left[ \exp(\boldsymbol{Q})\text{diag}(\boldsymbol{\omega}) \exp(-i\boldsymbol{K})^\top \right]$ as ①.

According to Euler's formula, we have:

$$
\begin{aligned}
① &= \text{Re}\left\{ (\cos \boldsymbol{Q} + i \sin \boldsymbol{Q})\text{diag}(\boldsymbol{\omega}) \left[ \cos(-\boldsymbol{K}^\top) + i \sin(-\boldsymbol{K}^\top) \right] \right\} \\
&= \text{Re}\left\{ (\cos \boldsymbol{Q} + i \sin \boldsymbol{Q})\text{diag}(\boldsymbol{\omega}) \left[ \cos(\boldsymbol{K}^\top) - i \sin(\boldsymbol{K}^\top) \right] \right\} \\
&= \cos \boldsymbol{Q}\,\text{diag}(\boldsymbol{\omega}) \cos(\boldsymbol{K}^\top) + \sin \boldsymbol{Q}\,\text{diag}(\boldsymbol{\omega}) \sin(\boldsymbol{K}^\top) \\
&= \cos\left( \boldsymbol{X}^\top \tilde{\boldsymbol{B}}^\top \right) \text{diag}(\boldsymbol{\omega}) \cos(\tilde{\boldsymbol{C}}\boldsymbol{X}) + \sin\left( \boldsymbol{X}^\top \tilde{\boldsymbol{B}}^T \right) \text{diag}(\boldsymbol{\omega}) \sin(\tilde{\boldsymbol{C}}\boldsymbol{X})
\end{aligned}
$$

$$
d\boldsymbol{G} = d[\sigma(①) \cdot \boldsymbol{V}] = [d\sigma(①)]\boldsymbol{V} + \sigma(①) \cdot d\boldsymbol{V}
$$

$$
d\boldsymbol{V} = d\left[ (\tilde{\boldsymbol{D}}\boldsymbol{X})^\top \right] = d\left( \boldsymbol{X}^\top \tilde{\boldsymbol{D}}^\top \right) = (d\boldsymbol{X})^\top \tilde{\boldsymbol{D}}^\top
$$

$$d\sigma(\mathbb{1}) = \sigma^{'}(\mathbb{1}) \odot d\mathbb{1}$$

$$d\mathbb{1} = d\left[\cos\left(\boldsymbol{X}^\top \tilde{\boldsymbol{B}}^\top\right)\mathrm{diag}(\boldsymbol{\omega})\cos\left(\tilde{\boldsymbol{C}}\boldsymbol{X}\right) + \sin\left(\boldsymbol{X}^\top \tilde{\boldsymbol{B}}^\top\right)\mathrm{diag}(\boldsymbol{\omega})\sin\left(\tilde{\boldsymbol{C}}\boldsymbol{X}\right)\right]$$

$$= d\left\{\left[\cos\left(\boldsymbol{X}^\top \tilde{\boldsymbol{B}}^\top\right)\mathrm{diag}(\boldsymbol{\omega})\right]\cdot\cos\left(\tilde{\boldsymbol{C}}\boldsymbol{X}\right)\right\} + d\left\{\left[\sin\left(\boldsymbol{X}^\top \tilde{\boldsymbol{B}}^\top\right)\mathrm{diag}(\boldsymbol{\omega})\right]\cdot\sin\left(\tilde{\boldsymbol{C}}\boldsymbol{X}\right)\right\}$$

$$= \left\{d\left[\cos\left(\boldsymbol{X}^\top \hat{\boldsymbol{B}}^\top\right)\mathrm{diag}(\boldsymbol{\omega})\right]\right\}\cdot\cos\left(\tilde{\boldsymbol{C}}\boldsymbol{X}\right) + \cos\left(\boldsymbol{X}^\top \hat{\boldsymbol{B}}^\top\right)\mathrm{diag}(\boldsymbol{\omega})\cdot d\cos\left(\tilde{\boldsymbol{C}}\boldsymbol{X}\right)$$

$$+ \left\{d\left[\sin\left(\boldsymbol{X}\tilde{\boldsymbol{B}}^\top\right)\mathrm{diag}(\boldsymbol{\omega})\right]\right\}\cdot\sin\left(\tilde{\boldsymbol{C}}\boldsymbol{X}\right) + \sin\left(\boldsymbol{X}^\top \tilde{\boldsymbol{B}}^\top\right)\mathrm{diag}(\boldsymbol{\omega})\cdot d\sin\left(\tilde{\boldsymbol{C}}\boldsymbol{X}\right)$$

$$= \left[d\cos\left(\boldsymbol{X}^\top \tilde{\boldsymbol{B}}^\top\right)\right]\cdot\mathrm{diag}(\boldsymbol{\omega})\cdot\cos\left(\tilde{\boldsymbol{C}}\boldsymbol{X}\right) + \cos\left(\boldsymbol{X}^\top \tilde{\boldsymbol{B}}^\top\right)\cdot\mathrm{diag}(\boldsymbol{\omega})\cdot\left[d\cos\left(\tilde{\boldsymbol{C}}\boldsymbol{X}\right)\right]$$

$$+ \left[d\sin\left(\boldsymbol{X}^\top \tilde{\boldsymbol{B}}^\top\right)\right]\cdot\mathrm{diag}(\boldsymbol{\omega})\cdot\sin\left(\tilde{\boldsymbol{C}}\boldsymbol{X}\right) + \sin\left(\boldsymbol{X}^\top \tilde{\boldsymbol{B}}^\top\right)\mathrm{diag}(\boldsymbol{\omega})\cdot\left[d\sin\left(\tilde{\boldsymbol{C}}\boldsymbol{X}\right)\right]$$

$$= \left[-\sin\left(\boldsymbol{X}^\top \tilde{\boldsymbol{B}}^\top\right)\odot d\left(\boldsymbol{X}^\top \tilde{\boldsymbol{B}}^\top\right)\right]\cdot\mathrm{diag}(\boldsymbol{\omega})\cdot\cos\left(\tilde{\boldsymbol{C}}\boldsymbol{X}\right) + \cos\left(\boldsymbol{X}^\top \tilde{\boldsymbol{B}}^\top\right)\cdot\mathrm{diag}(\boldsymbol{\omega})$$

$$\cdot\left[-\sin\left(\tilde{\boldsymbol{C}}\boldsymbol{X}\right)\odot d(\tilde{\boldsymbol{C}}\boldsymbol{X})\right] + \left[\cos\left(\boldsymbol{X}^\top \tilde{\boldsymbol{B}}^\top\right)\odot d\left(\boldsymbol{X}^\top \tilde{\boldsymbol{B}}^\top\right)\right]\cdot\mathrm{diag}(\boldsymbol{\omega})\cdot\sin\left(\tilde{\boldsymbol{C}}\boldsymbol{X}\right)$$

$$+ \sin\left(\boldsymbol{X}^\top \tilde{\boldsymbol{B}}^\top\right)\cdot\mathrm{diag}(\boldsymbol{\omega})\cdot\left[\cos\left(\tilde{\boldsymbol{C}}\boldsymbol{X}\right)\odot d(\tilde{\boldsymbol{C}}\boldsymbol{X})\right]$$

$$dy = \mathrm{tr}\left[\left(\frac{\partial y}{\partial \boldsymbol{G}}\right)^\top d\boldsymbol{G}\right]$$

$$= \mathrm{tr}\left(\left(\frac{\partial y}{\partial \boldsymbol{G}}\right)^\top\cdot\left\{\left[\sigma^{'}(\mathbb{1})\odot d\mathbb{1}\right]\boldsymbol{V} + \sigma(\mathbb{1})\cdot\left[(d\boldsymbol{X})^\top \tilde{\boldsymbol{D}}^\top\right]\right\}\right)$$

$$= \mathrm{tr}\left\{\left(\frac{\partial y}{\partial \boldsymbol{G}}\right)^\top\cdot\left[\sigma^{'}(\mathbb{1})\odot d\mathbb{1}\right]\cdot\boldsymbol{V}\right\} + \mathrm{tr}\left[\left(\frac{\partial y}{\partial \boldsymbol{G}}\right)^\top\cdot\sigma(\mathbb{1})\cdot(d\boldsymbol{X})^\top \tilde{\boldsymbol{D}}^\top\right]$$

$$\mathrm{tr}\left[\left(\frac{\partial y}{\partial \boldsymbol{G}}\right)^\top\cdot\sigma(\mathbb{1})\cdot(d\boldsymbol{X})^\top \tilde{\boldsymbol{D}}^\top\right] \qquad = \mathrm{tr}\left[\tilde{\boldsymbol{D}}^\top\left(\frac{\partial y}{\partial \boldsymbol{G}}\right)^\top\cdot\sigma(\mathbb{1})(d\boldsymbol{X})^\top\right]$$

For $\boldsymbol{A}$, since $\mathrm{tr}(\boldsymbol{A}) = \mathrm{tr}(\boldsymbol{A}^\top) \implies \quad = \mathrm{tr}\left\{(d\boldsymbol{X})\left[\tilde{\boldsymbol{D}}^\top\left(\frac{\partial y}{\partial \boldsymbol{G}}\right)^\top\sigma(\mathbb{1})\right]^\top\right\}$

For $\boldsymbol{A}, \boldsymbol{B}, \mathrm{tr}(\boldsymbol{A}\boldsymbol{B}) = \mathrm{tr}(\boldsymbol{B}\boldsymbol{A}) \implies \quad = \mathrm{tr}\left\{\left[\tilde{\boldsymbol{D}}^\top\left(\frac{\partial y}{\partial \boldsymbol{G}}\right)^\top\sigma(\mathbb{1})\right]^\top d\boldsymbol{X}\right\}$

Remark $\mathrm{tr}\left\{\left(\frac{\partial y}{\partial \boldsymbol{G}}\right)^\top\cdot\left[\sigma^{'}(\mathbb{1})\odot d\mathbb{1}\right]\cdot\boldsymbol{V}\right\} = \text{part1} + \text{part2} + \text{part3} + \text{part4}$

$$\text{part1} = \mathrm{tr}\left(\left(\frac{\partial y}{\partial \boldsymbol{G}}\right)^\top\left\{\sigma^{'}(\mathbb{1})\odot\left[\left[-\sin\left(\boldsymbol{X}^\top \tilde{\boldsymbol{B}}^\top\right)\odot d\left(\boldsymbol{X}^\top \tilde{\boldsymbol{B}}^\top\right)\right]\mathrm{diag}(\boldsymbol{\omega})\cos\left(\tilde{\boldsymbol{C}}\boldsymbol{X}\right)\right]\right\}\boldsymbol{V}\right)$$

$$= \mathrm{tr}\left(\boldsymbol{V}\left(\frac{\partial y}{\partial \boldsymbol{G}}\right)^\top\left\{\sigma^{'}(\mathbb{1})\odot\left[\left[-\sin\left(\boldsymbol{X}^\top \tilde{\boldsymbol{B}}^\top\right)\odot d\left(\boldsymbol{X}^\top \tilde{\boldsymbol{B}}^\top\right)\right]\mathrm{diag}(\boldsymbol{\omega})\cos\left(\tilde{\boldsymbol{C}}\boldsymbol{X}\right)\right]\right\}\right)$$

$$= \mathrm{tr}(\left\{\left[\left(\frac{\partial y}{\partial \boldsymbol{G}}\right)\boldsymbol{V}^\top\right]\odot\sigma^{'}(\mathbb{1})\right\}^\top$$

$$\cdot\left\{\left[-\sin\left(\boldsymbol{X}^\top \tilde{\boldsymbol{B}}^\top\right)\odot d\left(\boldsymbol{X}^\top \tilde{\boldsymbol{B}}^\top\right)\right]\cdot\mathrm{diag}(\boldsymbol{\omega})\cdot\cos\left(\tilde{\boldsymbol{C}}\boldsymbol{X}\right)\right\})$$

$$= \text{tr}(\text{diag}(\omega) \cdot \cos{(\tilde{\boldsymbol{C}}\boldsymbol{X})} \cdot \left\{ \left[ \left( \frac{\partial y}{\partial \boldsymbol{G}} \right) \boldsymbol{V}^\top \right] \odot \sigma^{'}(\mathbb{O}) \right\}^\top$$

$$\cdot \left[ -\sin{\left( \boldsymbol{X}^\top \tilde{\boldsymbol{B}}^\top \right)} \odot d \left( \boldsymbol{X}^\top \tilde{\boldsymbol{B}}^\top \right) \right])$$

$$= \text{tr}(\left\{ \left[ \left[ \left( \frac{\partial y}{\partial \boldsymbol{G}} \right) \boldsymbol{V}^\top \right] \odot \sigma^{'}(\mathbb{O})[\text{diag}(\boldsymbol{\omega}) \cdot \cos{(\tilde{\boldsymbol{C}}\boldsymbol{X})}]^\top \right] \odot \left[ -\sin{\left( \boldsymbol{X}^\top \tilde{\boldsymbol{B}}^\top \right)} \right] \right\}^\top$$

$$\cdot d \left( \boldsymbol{X}^\top \tilde{\boldsymbol{B}}^\top \right))$$

$$= -\text{tr}(\left\{ \left[ \left[ \left( \frac{\partial y}{\partial \boldsymbol{G}} \right) \boldsymbol{V}^\top \right] \odot \sigma^{'}(\mathbb{O}) \cdot [\text{diag}(\boldsymbol{\omega}) \cdot \cos{(\tilde{\boldsymbol{C}}\boldsymbol{X})}]^\top \right] \odot \sin{\left( \boldsymbol{X}^\top \tilde{\boldsymbol{B}}^\top \right)} \right\}^\top$$

$$\cdot (d\boldsymbol{X})^\top \tilde{\boldsymbol{B}}^\top)$$

Remark $\boldsymbol{F} = \left\{ \left[ \left[ \left( \frac{\partial y}{\partial \boldsymbol{G}} \right) \boldsymbol{V}^\top \right] \odot \sigma^{'}(\mathbb{O}) \cdot [\text{diag}(\boldsymbol{\omega}) \cdot \cos{(\tilde{\boldsymbol{C}}\boldsymbol{X})}]^\top \right] \odot \sin{\left( \boldsymbol{X}^\top \tilde{\boldsymbol{B}}^\top \right)} \right\}^\top$

So part1 $= -\text{tr}\left( F(d\boldsymbol{X})^\top \tilde{\boldsymbol{B}}^\top \right) = -\text{tr}\left( \tilde{\boldsymbol{B}}^\top \boldsymbol{F}(d\boldsymbol{X})^\top \right) = -\text{tr}\left( d\boldsymbol{X} \cdot \left( \tilde{\boldsymbol{B}}^\top \boldsymbol{F} \right)^\top \right)$

$$= -\text{tr}\left( \left( \tilde{\boldsymbol{B}}^\top \boldsymbol{F} \right)^\top d\boldsymbol{X} \right) = \text{tr}\left( -\left( \tilde{\boldsymbol{B}}^\top \boldsymbol{F} \right)^\top d\boldsymbol{X} \right)$$

part2 $= \text{tr}\left( \left( \frac{\partial y}{\partial \boldsymbol{G}} \right)^\top \cdot \left\{ \sigma^{'}(\mathbb{O}) \odot \left[ \cos{\left( \boldsymbol{X}^\top \tilde{\boldsymbol{B}}^\top \right)} \cdot \text{diag}(\boldsymbol{\omega}) \cdot \left[ -\sin{(\tilde{\boldsymbol{C}}\boldsymbol{X})} \odot d(\tilde{\boldsymbol{C}}\boldsymbol{X}) \right] \right] \right\} \cdot \boldsymbol{V} \right)$

$$= \text{tr}(\left\{ \left[ \left( \frac{\partial y}{\partial \boldsymbol{G}} \right) \boldsymbol{V}^\top \right] \odot \sigma^{'}(\mathbb{O}) \right\}^\top$$

$$\cdot \left\{ \cos{\left( \boldsymbol{X}^\top \tilde{\boldsymbol{B}}^\top \right)} \cdot \text{diag}(\boldsymbol{\omega}) \cdot \left[ -\sin{(\tilde{\boldsymbol{C}}\boldsymbol{X})} \odot d \left( \tilde{\boldsymbol{C}}\boldsymbol{X} \right) \right] \right\})$$

$$= \text{tr}(\left[ \left( \left\{ \left[ \left( \frac{\partial y}{\partial \boldsymbol{G}} \right) \boldsymbol{V}^\top \right] \odot \sigma^{'}(\mathbb{O}) \right\}^\top \cdot \cos{\left( \boldsymbol{X}^\top \tilde{\boldsymbol{B}}^\top \right)} \cdot \text{diag}(\boldsymbol{\omega}) \right)^\top \odot [-\sin{(\tilde{\boldsymbol{C}}\boldsymbol{X})}] \right]^\top$$

$$\cdot d(\tilde{\boldsymbol{C}}\boldsymbol{X}))$$

Remark $N = \left[ \left( \left\{ \left[ \left( \frac{\partial y}{\partial \boldsymbol{G}} \right) \boldsymbol{V}^\top \right] \odot \sigma^{'}(\mathbb{O}) \right\}^\top \cos{\left( \boldsymbol{X}^\top \tilde{\boldsymbol{B}}^\top \right)} \cdot \text{diag}(\boldsymbol{\omega}) \right)^\top \odot [-\sin{(\tilde{\boldsymbol{C}}\boldsymbol{X})}] \right]^\top$

So part2 $= \text{tr}(\boldsymbol{N}\tilde{\boldsymbol{C}}d\boldsymbol{X}) = \text{tr}\left( \left[ (\boldsymbol{N}\tilde{\boldsymbol{C}})^\top \right]^\top d\boldsymbol{X} \right) = \text{tr}\left[ \left( \tilde{\boldsymbol{C}}^\top \boldsymbol{N}^\top \right)^\top d\boldsymbol{X} \right]$

part3 $= \text{tr}\left( \boldsymbol{V} \left( \frac{\partial y}{\partial \boldsymbol{G}} \right)^\top \cdot \left\{ \sigma^{'}(\mathbb{O}) \odot \left[ \left[ \cos{\left( \boldsymbol{X}^\top \tilde{\boldsymbol{B}}^\top \right)} \odot d \left( \boldsymbol{X}^\top \tilde{\boldsymbol{B}}^\top \right) \right] \text{diag}(\boldsymbol{\omega}) \sin{(\tilde{\boldsymbol{C}}\boldsymbol{X})} \right] \right\} \right)$

$$= \text{tr}\left( \left[ \left[ \left( \frac{\partial y}{\partial \boldsymbol{G}} \right) \boldsymbol{V}^\top \right] \odot \sigma^{'}(\mathbb{O}) \right]^\top \cdot \left[ \cos{\left( \boldsymbol{X}^\top \tilde{\boldsymbol{B}}^\top \right)} \odot d \left( \boldsymbol{X}^\top \tilde{\boldsymbol{B}}^\top \right) \right] \text{diag}(\boldsymbol{\omega}) \sin{(\tilde{\boldsymbol{C}}\boldsymbol{X})} \right)$$

$$= \text{tr}\left( \text{diag}(\boldsymbol{\omega}) \sin{(\tilde{\boldsymbol{C}}\boldsymbol{X})} \left[ \left[ \left( \frac{\partial y}{\partial \boldsymbol{G}} \right) \boldsymbol{V}^\top \right] \odot \sigma^{'}(\mathbb{O}) \right]^\top \left[ \cos{\left( \boldsymbol{X}^\top \tilde{\boldsymbol{B}}^\top \right)} \odot d \left( \boldsymbol{X}^\top \tilde{\boldsymbol{B}}^\top \right) \right] \right)$$

$$= \text{tr}(\left[ \left[ \text{diag}(\boldsymbol{\omega}) \cdot \sin{(\tilde{\boldsymbol{C}}\boldsymbol{X})} \cdot \left[ \left[ \left( \frac{\partial y}{\partial \boldsymbol{G}} \right) \boldsymbol{V}^\top \right] \odot \sigma^{'}(\mathbb{O}) \right]^\top \right]^\top \odot \cos{\left( \boldsymbol{X}^\top \tilde{\boldsymbol{B}}^\top \right)} \right]^\top$$

$$\cdot d \left( \boldsymbol{X}^\top \tilde{\boldsymbol{B}}^\top \right))$$

Remark $R = \left[ \left[ \text{diag}(\boldsymbol{\omega}) \cdot \sin\left(\tilde{\boldsymbol{C}}\boldsymbol{X}\right) \cdot \left[ \left[ \left( \frac{\partial y}{\partial \boldsymbol{G}} \right) \boldsymbol{V}^\top \right] \odot \sigma'(\mathbb{1}) \right]^\top \right]^\top \odot \cos\left(\boldsymbol{X}^\top \tilde{\boldsymbol{B}}^\top\right) \right]^\top$

So part3 $= \text{tr}\left( \tilde{\boldsymbol{B}}^\top \boldsymbol{R}(d\boldsymbol{X})^\top \right) = \text{tr}\left( d\boldsymbol{X} \left( \tilde{\boldsymbol{B}}^\top \boldsymbol{R} \right)^\top \right) = \text{tr}\left( \left( \tilde{\boldsymbol{B}}^\top \boldsymbol{R} \right)^\top d\boldsymbol{X} \right)$

$\text{part4} = \text{tr}\left( \boldsymbol{V}\left( \frac{\partial y}{\partial \boldsymbol{G}} \right)^\top \cdot \left\{ \sigma'(\mathbb{1}) \odot \left[ \sin\left(\boldsymbol{X}^\top \tilde{\boldsymbol{B}}^\top\right) \cdot \text{diag}(\boldsymbol{\omega}) \cdot \left[ \cos\left(\tilde{\boldsymbol{C}}\boldsymbol{X}\right) \odot d(\tilde{\boldsymbol{C}}\boldsymbol{X}) \right] \right] \right\} \right)$

$= \text{tr}\left( \left\{ \left[ \left( \frac{\partial y}{\partial \boldsymbol{G}} \right) \boldsymbol{V}^\top \right] \odot \sigma'(\mathbb{1}) \right\}^\top \left\{ \sin\left(\boldsymbol{X}^\top \tilde{\boldsymbol{B}}^\top\right) \text{diag}(\boldsymbol{\omega}) \cdot \left[ \cos\left(\tilde{\boldsymbol{C}}\boldsymbol{X}\right) \odot d(\tilde{\boldsymbol{C}}\boldsymbol{X}) \right] \right\} \right)$

$= \text{tr}(\left\{ \left[ \left\{ \left[ \left( \frac{\partial y}{\partial \boldsymbol{G}} \right) \boldsymbol{V}^\top \right] \odot \sigma'(\mathbb{1}) \right\}^\top \cdot \sin\left(\boldsymbol{X}^\top \tilde{\boldsymbol{B}}^\top\right) \cdot \text{diag}(\boldsymbol{\omega}) \right]^\top \odot \cos\left(\tilde{\boldsymbol{C}}\boldsymbol{X}\right) \right\}^\top$

$\cdot \tilde{\boldsymbol{C}} d\boldsymbol{X})$

Remark $J = \left\{ \left[ \left\{ \left[ \left( \frac{\partial y}{\partial \boldsymbol{G}} \right) \boldsymbol{V}^\top \right] \odot \sigma'(\mathbb{1}) \right\}^\top \cdot \sin\left(\boldsymbol{X}^\top \tilde{\boldsymbol{B}}^\top\right) \cdot \text{diag}(\boldsymbol{\omega}) \right]^\top \odot \cos\left(\tilde{\boldsymbol{C}}\boldsymbol{X}\right) \right\}^\top$

So part4 $= \text{tr}\left( \left( \tilde{\boldsymbol{C}}^\top \boldsymbol{J}^\top \right)^\top d\boldsymbol{X} \right)$

$\left( \frac{\partial y}{\partial \boldsymbol{X}} \right)^\top = \left[ \tilde{\boldsymbol{D}}^\top \left( \frac{\partial y}{\partial \boldsymbol{G}} \right)^\top \sigma(\mathbb{1}) \right] + \left( -\tilde{\boldsymbol{B}}^\top \boldsymbol{F} \right) + \tilde{\boldsymbol{C}}^\top \boldsymbol{N}^\top + \tilde{\boldsymbol{B}}^\top \boldsymbol{R} + \tilde{\boldsymbol{C}}^\top \boldsymbol{J}^\top$

$\boldsymbol{F} = \left\{ \left[ \left[ \left( \frac{\partial y}{\partial \boldsymbol{G}} \right) \boldsymbol{V}^\top \right] \odot \sigma'(\mathbb{1}) \cdot \left[ \text{diag}(\boldsymbol{\omega}) \cdot \cos\left(\tilde{\boldsymbol{C}}\boldsymbol{X}\right) \right]^\top \right] \odot \sin\left(\boldsymbol{X}^\top \tilde{\boldsymbol{B}}^\top\right) \right\}^\top$

$= \left\{ \left[ \left( \frac{\partial y}{\partial \boldsymbol{G}} \right) \boldsymbol{V}^\top \right] \odot \sigma'(\mathbb{1}) \cdot \left[ \text{diag}(\boldsymbol{\omega}) \cdot \cos\left(\tilde{\boldsymbol{C}}\boldsymbol{X}\right) \right]^\top \right\}^\top \odot \sin\left(\tilde{\boldsymbol{B}}\boldsymbol{X}\right)$

$= \left\{ \text{diag}(\boldsymbol{\omega}) \cdot \cos\left(\tilde{\boldsymbol{C}}\boldsymbol{X}\right) \cdot \left[ \left[ \left( \frac{\partial y}{\partial \boldsymbol{G}} \right) \boldsymbol{V}^\top \right] \odot \sigma'(\mathbb{1}) \right]^\top \right\} \odot \sin\left(\boldsymbol{Q}^\top\right)$

$= \left\{ \text{diag}(\boldsymbol{\omega}) \cdot \cos\left(\boldsymbol{K}^\top\right) \cdot \left( \left[ \boldsymbol{V}\left( \frac{\partial y}{\partial \boldsymbol{G}} \right)^\top \right] \odot \left[ \sigma'(\mathbb{1}) \right]^\top \right) \right\} \odot \sin\left(\boldsymbol{Q}^\top\right)$

$= \{ \text{diag}(\boldsymbol{\omega}) \text{Re}\left( \exp\left( i\boldsymbol{K}^\top \right) \right)$

$\cdot \left[ \left[ \boldsymbol{V}\left( \frac{\partial y}{\partial \boldsymbol{G}} \right)^\top \right] \odot \sigma'\left( \text{Re}\left[ \exp\left( -i\boldsymbol{K} \right) \text{diag}(\boldsymbol{\omega}) \exp\left( i\boldsymbol{Q}^\top \right) \right] \right) \right] \} \odot \text{Im}\left[ \exp\left( i\boldsymbol{Q}^\top \right) \right]$

$\boldsymbol{N}^\top = \left\{ \left[ \left[ \left( \frac{\partial y}{\partial \boldsymbol{G}} \right) \boldsymbol{V}^\top \right] \odot \sigma'(\mathbb{1}) \right]^\top \cdot \cos\left(\boldsymbol{X}^\top \tilde{\boldsymbol{B}}^\top\right) \cdot \text{diag}(\boldsymbol{\omega}) \right\}^\top \odot \left[ -\sin\left(\tilde{\boldsymbol{C}}\boldsymbol{X}\right) \right]$

$= \left\{ \left[ \left[ \boldsymbol{V}\left( \frac{\partial y}{\partial \boldsymbol{G}} \right)^\top \right] \odot \sigma'\left( \mathbb{1}^\top \right) \right] \cdot \cos\left(\boldsymbol{X}^\top \tilde{\boldsymbol{B}}^\top\right) \cdot \text{diag}(\boldsymbol{\omega}) \right\}^\top \cdot \left[ -\sin\left(\boldsymbol{K}^\top\right) \right]$

$= \left\{ \text{diag}(\boldsymbol{\omega}) \cdot \cos\left(\tilde{\boldsymbol{B}}\boldsymbol{X}\right) \cdot \left[ \left[ \left( \frac{\partial y}{\partial \boldsymbol{G}} \right) \boldsymbol{V}^\top \right] \odot \sigma'(\mathbb{1}) \right] \right\} \odot \left[ -\sin\left(\boldsymbol{K}^\top\right) \right]$

$= \{ \text{diag}(\boldsymbol{\omega}) \text{Re}\left[ \exp\left( i\boldsymbol{Q}^\top \right) \right]$

$\cdot \left[ \left[ \left( \frac{\partial y}{\partial \boldsymbol{G}} \right)^\top \boldsymbol{V}^\top \right] \odot \sigma'\left( \text{Re}\left[ \exp\left( i\boldsymbol{Q} \right) \text{diag}(\boldsymbol{\omega}) \exp\left( -i\boldsymbol{K} \right)^\top \right] \right) \right] \} \odot \left\{ -\text{Im}\left[ \exp\left( i\boldsymbol{K}^\top \right) \right] \right\}$

$$\boldsymbol{R} = \left\{ \left[ \operatorname{diag}(\boldsymbol{\omega}) \sin\left(\tilde{\boldsymbol{C}}\boldsymbol{X}\right) \cdot \left[ \left[ \left( \frac{\partial y}{\partial \boldsymbol{G}} \right) \boldsymbol{V}^\top \right] \odot \sigma'(\mathbb{1}) \right]^\top \right]^\top \odot \cos\left(\boldsymbol{X}^\top \tilde{\boldsymbol{B}}^\top\right) \right\}^\top$$

$$= \left\{ \left[ \operatorname{diag}(\boldsymbol{\omega}) \sin\left(\boldsymbol{K}^\top\right) \left[ \left[ \boldsymbol{V}\left( \frac{\partial y}{\partial \boldsymbol{G}} \right)^\top \right] \odot \sigma'(\mathbb{1}^\top) \right] \right]^\top \odot \cos\left(\boldsymbol{X}^\top \tilde{\boldsymbol{B}}^\top\right) \right\}^\top$$

$$= \left[ \operatorname{diag}(\boldsymbol{\omega}) \sin\left(\boldsymbol{K}^\top\right) \left[ \left[ \boldsymbol{V}\left( \frac{\partial y}{\partial \boldsymbol{G}} \right)^\top \right] \odot \sigma'(\mathbb{1}^\top) \right] \right] \odot \cos\left(\tilde{\boldsymbol{B}}\boldsymbol{X}\right)$$

$$= \left\{ \operatorname{diag}(\boldsymbol{\omega})\operatorname{Im}\left[ \exp\left(i\boldsymbol{K}^\top\right) \right] \right.$$

$$\left. \cdot \left[ \left[ \boldsymbol{V}\left( \frac{\partial y}{\partial \boldsymbol{G}} \right)^\top \right] \odot \sigma'\left( \operatorname{Re}\left[ \exp\left( -i\boldsymbol{K}\right)\operatorname{diag}(\boldsymbol{\omega})\exp\left( i\boldsymbol{Q}^\top\right) \right] \right) \right] \right\} \odot \operatorname{Re}\left[ \exp\left( i\boldsymbol{Q}^\top\right) \right]$$

$$\boldsymbol{J}^\top = \left[ \left[ \left[ \left( \frac{\partial y}{\partial \boldsymbol{G}} \right) \boldsymbol{V}^\top \right] \odot \sigma'(\mathbb{1}) \right]^\top \cdot \sin\left(\boldsymbol{X}^\top \tilde{\boldsymbol{B}}^\top\right) \cdot \operatorname{diag}(\boldsymbol{\omega}) \right]^\top \odot \cos\left(\tilde{\boldsymbol{C}}\boldsymbol{X}\right)$$

$$= \left\{ \operatorname{diag}(\boldsymbol{\omega}) \sin\left(\tilde{\boldsymbol{B}}\boldsymbol{X}\right) \left[ \left[ \left( \frac{\partial y}{\partial \boldsymbol{G}} \right) \boldsymbol{V}^\top \right] \odot \sigma'(\mathbb{1}) \right] \right\} \odot \cos\left(\boldsymbol{K}^\top\right)$$

$$= \left\{ \operatorname{diag}(\boldsymbol{\omega})\operatorname{Im}\left[ \exp\left(i\boldsymbol{Q}^\top\right) \right] \right.$$

$$\left. \cdot \left[ \left[ \left( \frac{\partial y}{\partial \boldsymbol{G}} \right) \boldsymbol{V}^\top \right] \odot \sigma'\left( \operatorname{Re}\left[ \exp\left( i\boldsymbol{Q}\right)\operatorname{diag}(\boldsymbol{\omega})\exp\left( -i\boldsymbol{K}\right)^\top \right] \right) \right] \right\} \odot \operatorname{Re}\left[ \exp\left( i\boldsymbol{K}^\top\right) \right]$$

$$\frac{\partial y}{\partial \boldsymbol{X}} = \tilde{\boldsymbol{D}}^\top \left( \frac{\partial y}{\partial \boldsymbol{G}} \right)^\top \sigma\left( \operatorname{Re}\left[ \exp\left( i\boldsymbol{Q}\right)\operatorname{diag}(\boldsymbol{\omega})\exp\left( -i\boldsymbol{K}\right)^\top \right] \right)$$

$$- \tilde{\boldsymbol{B}}^\top \left( \left\{ \operatorname{diag}(\boldsymbol{\omega}) \cdot \operatorname{Re}\left[ \exp\left( i\boldsymbol{K}^\top\right) \right] \right. \right.$$

$$\left. \left. \cdot \left[ \left[ \boldsymbol{V}\left( \frac{\partial y}{\partial \boldsymbol{G}} \right)^\top \right] \odot \sigma'\left( \operatorname{Re}\left[ \exp\left( -i\boldsymbol{K}\right)\operatorname{diag}(\boldsymbol{\omega})\exp\left( i\boldsymbol{Q}^\top\right) \right] \right) \right] \right\} \odot \operatorname{Im}\left[ \exp\left( i\boldsymbol{Q}^\top\right) \right] \right)$$

$$- \tilde{\boldsymbol{C}}^\top \left( \left\{ \operatorname{diag}(\boldsymbol{\omega}) \cdot \operatorname{Re}\left[ \exp\left( i\boldsymbol{Q}^\top\right) \right] \right. \right.$$

$$\left. \left. \cdot \left[ \left[ \left( \frac{\partial y}{\partial \boldsymbol{G}} \right) \boldsymbol{V}^\top \right] \odot \sigma'\left( \operatorname{Re}\left[ \exp\left( i\boldsymbol{Q}\right)\operatorname{diag}(\boldsymbol{\omega})\exp\left( -i\boldsymbol{K}\right)^\top \right] \right) \right] \right\} \odot \operatorname{Im}\left[ \exp\left( i\boldsymbol{K}^\top\right) \right] \right)$$

$$+ \tilde{\boldsymbol{B}}^\top \left( \left\{ \operatorname{diag}(\boldsymbol{\omega}) \cdot \operatorname{Im}\left[ \exp\left( i\boldsymbol{K}^\top\right) \right] \right. \right.$$

$$\left. \left. \cdot \left[ \left[ \boldsymbol{V}\left( \frac{\partial y}{\partial \boldsymbol{G}} \right)^\top \right] \odot \sigma'\left( \operatorname{Re}\left[ \exp\left( -i\boldsymbol{K}\right)\operatorname{diag}(\boldsymbol{\omega})\exp\left( i\boldsymbol{Q}^\top\right) \right] \right) \right] \right\} \odot \operatorname{Re}\left[ \exp\left( i\boldsymbol{Q}^\top\right) \right] \right)$$

$$+ \tilde{\boldsymbol{C}}^\top \left( \left\{ \operatorname{diag}(\boldsymbol{\omega}) \cdot \operatorname{Im}\left[ \exp\left( i\boldsymbol{Q}^\top\right) \right] \right. \right.$$

$$\left. \left. \cdot \left[ \left[ \left( \frac{\partial y}{\partial \boldsymbol{G}} \right) \boldsymbol{V}^\top \right] \odot \sigma'\left( \operatorname{Re}\left[ \exp\left( i\boldsymbol{Q}\right)\operatorname{diag}(\boldsymbol{\omega})\exp\left( -i\boldsymbol{K}\right)^\top \right] \right) \right] \right\} \odot \operatorname{Re}\left[ \exp\left( i\boldsymbol{K}^\top\right) \right] \right)$$

Correspondingly, we remark the equation as: $\frac{\partial y}{\partial \boldsymbol{X}} = \text{PART}I - \text{PART}II - \text{PART}III + \text{PART}IV + \text{PART}V$.

For a matrix $A$, we define the infinite norms of matrices as: $\parallel \boldsymbol{A} \parallel_\infty = \max\limits_i \sum_{j=1}^n |a_{ij}|$.

Since we can learn $\tilde{\boldsymbol{B}}, \tilde{\boldsymbol{C}}, \tilde{\boldsymbol{D}}, \operatorname{diag}(\boldsymbol{\omega}), \boldsymbol{X}$ and $\frac{\partial y}{\partial \boldsymbol{G}}$, we assume:

$\parallel \tilde{\boldsymbol{B}}^\top \parallel_\infty < \alpha, \parallel \tilde{\boldsymbol{C}}^\top \parallel_\infty < \beta, \parallel \operatorname{diag}(\boldsymbol{\omega}) \parallel_\infty < \zeta$,

$\parallel \tilde{\boldsymbol{D}}^\top \parallel_\infty < \gamma_1, \parallel \tilde{\boldsymbol{D}} \parallel_\infty < \gamma_2$, let $\gamma = \max\limits_i \{\gamma_1, \gamma_2\}$, then $\parallel \tilde{\boldsymbol{D}}^\top \parallel_\infty < \gamma, \parallel \tilde{\boldsymbol{D}} \parallel_\infty < \gamma$,

Similarily, $\parallel (\frac{\partial y}{\partial \boldsymbol{G}})^\top \parallel_\infty < \theta, \parallel (\frac{\partial y}{\partial \boldsymbol{G}}) \parallel_\infty < \theta$.

Note that each row of $\boldsymbol{X}$ can have at most one non-zero element due to the inherent sparsity of $\boldsymbol{X}$. We suppose the absolute value of every element in $\boldsymbol{X}$ is smaller than $\eta$, so we have $\| \boldsymbol{X} \|_\infty < \eta$ and $\| \boldsymbol{X}^\top \|_\infty < \mathrm{d}\eta$. According to the compatibility of this norm:

$$\mathrm{PART}I = \left\| \tilde{\boldsymbol{D}}^\top \left( \frac{\partial y}{\partial \boldsymbol{G}} \right)^\top \sigma \left( Re \left[ \exp\left( i\boldsymbol{Q} \right) \mathrm{diag}(\boldsymbol{\omega}) \exp\left( -i\boldsymbol{K} \right)^T \right] \right) \right\|_\infty$$

$$\leq \left\| \tilde{\boldsymbol{D}}^\top \right\|_\infty \cdot \| \left( \frac{\partial y}{\partial \boldsymbol{G}} \right)^\top \|_\infty \cdot \| \sigma \left( Re \left[ \exp\left( i\boldsymbol{Q} \right) \mathrm{diag}(\boldsymbol{\omega}) \exp\left( -i\boldsymbol{K} \right)^\top \right] \right) \|_\infty$$

$$\leq \gamma \cdot \theta \cdot \mathrm{m}$$

$$\mathrm{PART}II \leq \left\| \boldsymbol{B}^\top \right\|_\infty \| \mathrm{diag}\left( \boldsymbol{\omega} \right) \|_\infty \| Re \left[ \exp\left( i\boldsymbol{K}^T \right) \right] \|_\infty \| \boldsymbol{V} \|_\infty \left\| \left( \frac{\partial y}{\partial \boldsymbol{G}} \right)^\top \right\|_\infty \frac{1}{4}$$

$$\leq \alpha \cdot \zeta \cdot \mathrm{m} \cdot \mathrm{d} \cdot \eta \cdot \gamma \cdot \theta \cdot \frac{1}{4} = \frac{\alpha \gamma \eta \theta \zeta \mathrm{dm}}{4}$$

Similarily, $\mathrm{PART}III \leq \frac{\beta \zeta \gamma \eta \theta \mathrm{m}}{4}, \mathrm{PART}IV \leq \frac{\alpha \zeta \gamma \eta \theta \mathrm{dm}}{4}, \mathrm{PART}V \leq \frac{\beta \zeta \theta \gamma \eta \mathrm{m}}{4}$.

$$\left\| \frac{\partial y}{\partial \boldsymbol{X}} \right\|_\infty \leq \gamma \theta \mathrm{m} + \frac{\alpha \zeta \eta \gamma \theta \mathrm{dm}}{4} + \frac{\beta \zeta \gamma \eta \theta \mathrm{m}}{4} + \frac{\alpha \zeta \eta \gamma \theta \mathrm{dm}}{4} + \frac{\beta \zeta \gamma \eta \theta \mathrm{m}}{4}$$

$$= \gamma \theta \mathrm{m} + \frac{\alpha \zeta \eta \gamma \theta \mathrm{dm}}{2} + \frac{\beta \zeta \gamma \eta \theta \mathrm{m}}{2}$$

Remark $\gamma \theta + \frac{\alpha \zeta \eta \gamma \theta \mathrm{d}}{2} + \frac{\beta \zeta \gamma \eta \theta}{2} = C_1$, thus, $\left\| \frac{\partial y}{\partial \boldsymbol{X}} \right\|_\infty \leq C_1 \mathrm{m}$.

Note that $C_1$ is independent of the field number $m$. It can be seen that under certain regularity conditions, the gradient terms grow at most linearly with $m$.

For the traditional feature interaction algorithms, their gradients can be formulated as:

$$\boldsymbol{g} = \boldsymbol{X}_1^{\alpha_1} \odot \boldsymbol{X}_2^{\alpha_2} \odot \cdots \odot \boldsymbol{X}_m^{\alpha_m}$$

$$\frac{\partial \boldsymbol{g}}{\partial \boldsymbol{X}_i} = \boldsymbol{X}_1^{\alpha_1} \odot \boldsymbol{X}_2^{\alpha_2} \odot \cdots \odot \alpha_i \boldsymbol{X}_i^{\alpha_i - 1} \odot \boldsymbol{X}_{i+1}^{\alpha_{i+1}} \odot \cdots \odot \boldsymbol{X}_m^{\alpha_m} \;\;, \;\; y = f(\boldsymbol{g}).$$

We suppose there exists $j$, for all $i$ we all have $X_{ij} \geq M - \varepsilon$, thus:

$$\left| \frac{\partial \boldsymbol{g}}{\partial X_{ij}} \right| \geq \alpha_i \cdot (M - \varepsilon)^{\sum_{i=1}^m \alpha_i - 1}$$

$$\left| \frac{\partial y}{\partial X_{ij}} \right| \geq \alpha_i \cdot (M - \varepsilon)^{\sum_{i=1}^m \alpha_i - 1} \left\| \frac{\partial y}{\partial \boldsymbol{g}} \right\|_\infty$$

Let $t = \min_i \{\alpha_i\}$, thus we have:

$$\left| \frac{\partial \boldsymbol{g}}{\partial X_{ij}} \right| \geq \alpha_i \cdot (M - \varepsilon)^{mt - 1} \left\| \frac{\partial y}{\partial \boldsymbol{g}} \right\|_\infty$$

We can clearly see that the gradient terms of traditional feature interaction algorithms exponentially grow with the field number $m$.

# B    DATASETS

We evaluate RFM with five real-world classification datasets on representative tasks, including app recommendation (Frappe[2]), movie recommendation (MovieLens-1M[3], MovieLens-Tag[4]), click-through prediction (Criteo[5], Avazu[6]).

• The Criteo dataset is recognized as a prominent benchmark in the domain of Click-Through Rate (CTR) prediction, encompassing user logs over a span of seven days. It exhibits a balanced distribution of labels, maintaining a positive to negative ratio of approximately 1:3. The pre-processing approach adopted for managing this dataset can be found in EulerNet Tian et al. (2023).

• Avazu was utilized in the Avazu Click-Through Rate (CTR) prediction challenge, aiming to estimate the likelihood of a mobile advertisement being clicked. The Avazu dataset presents a positive to negative ratio of approximately 1:5. For preprocessing the dataset, the method delineated in EulerNet Tian et al. (2023) was adopted.

• ML-1M dataset is widely recognized as a prominent choice in the realm of recommendation systems research. Each training instance consists of a triplet of features representing users, movies, and ratings. Following the approach in EulerNet Tian et al. (2023), ratings of 1 and 2 are transformed to 0, ratings of 4 and 5 are converted to 1, and ratings of 3 are excluded. The dataset includes 7 categorical fields without multiple values, which are utilized and represented using embeddings.

• ML-Tag encompasses movie tagging data recorded by users across different time spans. Building on the approach by the work Cheng et al. (2020), our emphasis lies on tailoring tag recommendations to individual users. To achieve this, we structure the dataset in the (user_id, movie_id, tag_id) format.

• Frappe serves as a practical application recommendation dataset, featuring a context-aware log of app usage. It generates two negative tuples for each positive app usage log. The objective is to forecast app usage based on the context of usage, encompassing 10 semantic attributes like previous app usage count, weather, time, location, and more. To preprocess the dataset, we adopt the approach outlined in the work Cheng et al. (2020).

# C    BASELINES

We consider the following baseline methods for performance comparison:

***First-Order***:

• LR Richardson et al. (2007) utilizes the original field features as input for prediction, merely combining these features using corresponding weights.

***Second-Order***:

• FwFM Pan et al. (2018) takes into account the semantic significance among distinct feature fields and introduces a scalar weight to eliminate insignificant feature interactions.

• FmFM Sun et al. (2021) enhances FwFM by substituting the single scalar field weight with a matrix, and it computes the kernel product on the feature embeddings to capture significant feature interdependencies.

***High-Order***:

• NFM He & Chua (2017) NFM aggregates the result of the element-wise multiplication of input feature vectors, which is then processed through fully connected layers.

• CIN Lian et al. (2018) generates high-order cross features through the computation of outer products of feature vectors across various orders.

---

[2]https://www.baltrunas.info/research-menu/frappe
[3]https://grouplens.org/datasets/movielens/
[4]https://grouplens.org/datasets/movielens/
[5]https://labs.criteo.com/2014/02/kaggle-display-advertising-challenge-dataset/
[6]https://www.kaggle.com/c/avazu-ctr-prediction

- CrossNet Wang et al. (2021) models feature interactions explicitly through the calculation of the kernel product of input feature vectors.

- PNN Qu et al. (2016) capture feature interactions by combining inner or outer products of input feature vectors in a pairwise manner.

*Ensemble*:

- AutoInt Song et al. (2019) utilizes Multi-head Self-Attention to autonomously construct high-order characteristics. It stands as the pioneering endeavor to utilize Transformers for acquiring high-order feature interplays.

- DeepFM Guo et al. (2017) integrates classical factorization machines with a multilayer perceptron (MLP) to improve the modeling of high-order feature interactions.

- xDeepFM Lian et al. (2018) integrates the CIN model with an MLP.

- DCNV2 Wang et al. (2021) integrates the CrossNet model with an MLP.

*Adaptive-Order*:

- AFN Cheng et al. (2020) transforms features into a logarithmic space to flexibly grasp arbitrary-order feature interactions. The AFN+ enhancement involves the utilization of an MLP to enhance the underlying model.

- ARM-Net Cai et al. (2021) introduces a gated attention mechanism that adapts to instances to dynamically learn the orders of feature interactions. On the other hand, ARM-Net+ enhances the underlying model by incorporating an MLP.

- EulerNet Tian et al. (2023) employs Euler's formula to capture arbitrary-order feature interactions in the complex vector space, thus overcoming the non-negativity constraints present in the AFN.

These models compared in our experiments encompass various forms of feature interaction techniques. LR, as the most straightforward approach, utilizes feature weights for direct prediction development. FmFM and FwFM are relatively simple models that capture only second-order feature interactions. NFM, CIN, CrossNet, and PNN have the capacity to model higher-order feature interactions. AutoInt+, DeepFM, xDeepFM, and DCNV2 are ensemble methods that incorporate an MLP to enhance high-order feature interactions. AFN+, ARM-Net+, and EulerNet have the capacity to learn adaptive-order feature interactions.

## D   IMPLEMENTATION DETAILS

We reuse the baseline models and implement our models based on RecBole (Zhao et al., 2021; 2022; Xu et al., 2023), an open-source library[7]. For each method, extensive grid search is applied to find the optimal settings. Our evaluation follows the same experimental settings as EulerNet (Tian et al., 2023), by setting the size feature embedding to 16, and batch size to 1024. We set the learning rate from 1e-1 to 1e-4 on a log scale and then narrowed down to 5e-4 on a linear scale. The regularization parameter $\lambda$ is in $\{$1e-3, 1e-5, 1e-7$\}$. The optimizer is Adam (Kingma & Ba, 2014). For RFM, the number of self-attentive rotation layers is in $\{1, 2, 3\}$, the number of attention heads is in $\{1, 2, 4, 8\}$, and attention dimension is in $\{16, 32, 48, 64, 80\}$. The architecture of the amplification network is in $\{48, 128, 256 \times 256\}$. The hidden dimension of the group normalization is in $\{2, 4, 8, 16\}$. **We have provided our source code in the supplementary materials.**

Next, we detail the hyperparameters of each model, with the search space defined based on prior research Wang et al. (2021); Tian et al. (2023). For each baseline method, the MLP component's hidden size is selected from $\{64, 128, 256, 512\}$, the layer count from $\{1, 2, 3\}$, and the dropout rate from $\{0.0, 0.1, 0.2, 0.3, 0.4\}$. In the case of FwFM and FmFM, we employ field-wise linear weights. CIN and xDeepFM have layer sizes in $\{100, 200\}$, depth in $\{2, 3, 4\}$, identity activation, and direct or indirect computation. CrossNet and DCNV2 vary in cross-layer numbers from 1 to 4. Regarding PNN, we explore IPNN, OPNN, and different kernel types such as full matrix, vector, and number. AutoInt (Transformer) involves attention layer counts of 2 to 4, attention embedding sizes of $\{20, 32, 40\}$, attention head numbers of 2 to 3. For AFN, logarithmic neuron counts span

---

[7]https://recbole.io/

$\{40, 400, 800, 1000\}$. ARM-Net incorporates $\alpha$ sparsity values in $[1.0, 2.0, 3.0]$, attention head numbers in $\{1, 2, 4, 8\}$, and exponential neurons per head in $\{8, 16, 32, 64\}$. EulerNet experiments with Euler interaction layer counts in $\{1, 2, 3, 4\}$ and order vector numbers in $\{10, 20, 30, 40\}$.

## E COMPLEXITY ANALYSIS

For ease of analysis, we assume that the hidden size of different components is set to the same number. Let $m$ denote the number of feature fields, $h$ denote the head number, $d$ denote the embedding dimension, $d'$ denote the total attention dimension, $d_h$ denote the attention dimension of a single head, and $T$ denote the MLP hidden size of the amplification network.

**Time Complexity.** Within each self-attentive rotation layer, calculating attention weights for one head takes $\mathcal{O}(mdd_h + m^2 d_h)$ time. As for multi-head rotation, we use $d_h = d'/h$. Because we have $h$ heads, it takes $\mathcal{O}(mdd' + m^2 d')$ time altogether. The time complexity of a $N$-layer network is $\mathcal{O}(mNdd' + m^2 Nd')$. As for an $L$-layer amplification network, the time complexity is $\mathcal{O}(md'T + LT^2)$. Therefore, the time complexity of the RFM is of the same order as the Transformer.

**Space Complexity.** The embedding layer, which is a shared component in neural network-based methods, contains $nd$ parameters, where $n$ is the dimension of sparse representation of input feature and $d$ is the embedding size. As a self-attentive rotation layer contains the following weight: $\{\boldsymbol{W}_j^Q, \boldsymbol{W}_j^K, \boldsymbol{W}_j^V\}_{j=1}^m \in \mathbb{R}^{d \times d'}$, and $\boldsymbol{w} \in \mathbb{R}^{d'}$. In the multi-head rotation, since we follow the implementation of Transformer, which sets $d_h = d'/h$ and conducts the split operation to implement the head projection matrix (*i.e.*, $H_j^Q$ in Eq. 11), the total parameter number is equal to that of the single-head case. Due to the reduced dimension of each head, the total computational cost is similar to that of a single-head attention with full dimensionality. The space complexity of a $N$-layer network is $\mathcal{O}(mNdd')$. As for an $L$-layer amplification network, there are $\mathcal{O}(md'T + LT^2)$ parameters.

## F HYPER-PARAMETER STUDY

We study how the hyper-parameters impact the performance of RFM. We mainly focus on three hyper-parameters: the attention dimension, the number of attention heads and the number of attention layers.

**Influence of Different Attention Dimensions.** We investigate the performance with respect to the attention dimension $d'$ in the self-attentive rotation layer. As shown in Figure 5, on the Criteo and Avazu datasets, we can see that the performance increases as the attention dimension increases from 16 to 32. Whereas, on the Frappe dataset, RFM achieves the best performance as the attention dimension increases to 48. Continuously increasing the attention dimension does not yield a sustained improvement in model performance. The reason is that the model overfits when too many parameters are incorporated.

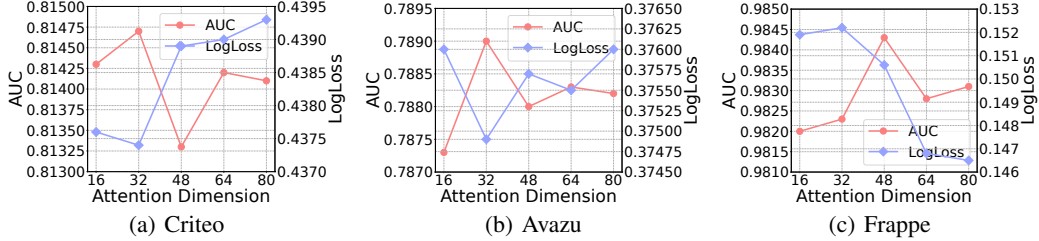

Figure 5: The performance w.r.t. the attention dimension $d'$.

**Influence of Different Attention Heads.** As mentioned in Section 3.1.2, the attention heads number $h$ controls the number of feature interaction terms. As shown in Figure 6, we can see that the performance increases as the attention head number increases from 2 to 4 on the Criteo and Avazu

datasets, showing the effectiveness of incorporating more feature interactions. The results are different on the Frappe dataset; the model performance varies significantly across different attention head numbers. The reason is that this data set is small, introducing too many interaction terms may introduce irrelevant noise that hurts the model performance.

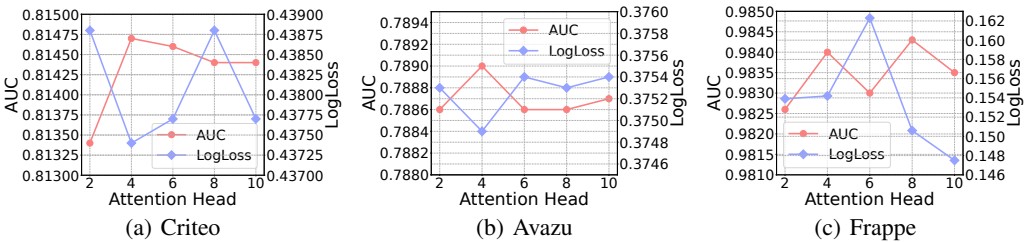

Figure 6: The performance w.r.t. the attention head number $h$.

**Influence of Different Attention Layer Number.** RFM is designed by stacking $L$ self-attentive rotation layers. To analyze the influence of $L$, we vary $L$ in the range of 1 to 5 to report the results in Figure 7. We can observe that the performance of RFM increases with the attention layer number at the beginning. However, model performance degrades when the attention layer number is set greater than 2 on the Criteo and Avazu dataset, whereas RFM achieves the best performance with a single layer. In practice, the layer number of RFM is usually set to 1 or 2, thereby ensuring the efficiency of our approach.

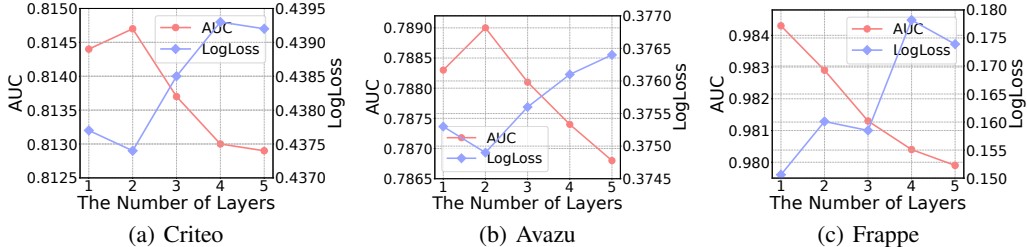

Figure 7: The performance w.r.t. the attention head number $h$.

# G   MORE ABLATION STUDIES

In this section, we conduct ablation studies to investigate the effectiveness of other components in RFM. The results are presented in Table 6.

**Projection Matrices.** As mentioned in Section 3.1.2, we employ a set of field-specific projection matrices (*i.e.,* $\{\boldsymbol{W}_j^Q, \boldsymbol{W}_j^K, \boldsymbol{W}_j^V \in \mathbb{R}^{d' \times d}\}_{j=1}^m$) to map the original feature embeddings into a set of queries, keys and values (*i.e.,* $\boldsymbol{Q}, \boldsymbol{K}, \boldsymbol{V}$). To verify its effectiveness, we compare it with the mapping approach of traditional transformers, *i.e.,* all fields use **shared matrices** $\boldsymbol{W}^Q, \boldsymbol{W}^K, \boldsymbol{W}^V$. We can observe that the model performance has a decrease when a shared projection matrix is incorporated for mapping the features from all fields. It demonstrates that our proposed approach is more suitable for capturing the field-specific semantics that improve the model's capacities.

**Activation Function.** Our proposed self-attentive rotation mechanism adopts the sigmoid as the activation function to quantify the feature relationships. It can be seen that the performance drops when replacing the sigmoid function with other commonly used activation functions (*i.e.,* softmax, ReLU and Tanh). The sigmoid function squashes the orders into a range between 0 and 1 without additional constraints (*e.g.,* the orders add up to 1 in softmax function). Therefore, the sigmoid function is more suitable for quantifying the relationships and capturing the useful feature interactions.

**Amplification Network.** As introduced in Section 3.2, RFM feeds the real and imaginary parts of the complex features into a shared MLP for enhancing the representations. Our aim is to ensure the consistency of complex vector operations, *i.e.,* the real and imaginary parts of a complex vector should have the same weights (*e.g.,* $\boldsymbol{W}(\boldsymbol{r} + i\boldsymbol{p}) = \boldsymbol{W}\boldsymbol{r} + i\boldsymbol{W}\boldsymbol{p}$). To verify its effectiveness, the variant "Splited MLP" feeds the real and imaginary vectors into two different MLPs which are independently learned during training. We can see that the model performance decreases when using splited MLPs. It shows that the consistency of complex vector operations has a large impact on the performance. Meanwhile, the shared architecture also improves the efficiency of our approach.

Table 6: More ablation study results. 'LL' denotes the LogLoss

| Models | Criteo | | Avazu | | ML-1M | | ML-Tag | | Frappe | |
|---|---|---|---|---|---|---|---|---|---|---|
| | AUC | LL | AUC | LL | AUC | LL | AUC | LL | AUC | LL |
| Base RFM | 0.8147 | 0.4374 | 0.7890 | 0.3749 | 0.9026 | 0.3090 | 0.9667 | 0.2049 | 0.9843 | 0.1506 |
| Shared matrices | 0.8138 | 0.4381 | 0.7877 | 0.3761 | 0.8997 | 0.3130 | 0.9661 | 0.2063 | 0.9825 | 0.1595 |
| Softmax | 0.8142 | 0.4381 | 0.7886 | 0.3754 | 0.8927 | 0.3249 | 0.9653 | 0.2076 | 0.9836 | 0.1537 |
| ReLU | 0.8141 | 0.4383 | 0.7887 | 0.3752 | 0.8972 | 0.3148 | 0.9641 | 0.2183 | 0.9838 | 0.1473 |
| Tanh | 0.8139 | 0.4384 | 0.7882 | 0.3754 | 0.9011 | 0.3123 | 0.9657 | 0.2091 | 0.9831 | 0.1603 |
| Splited MLP | 0.8139 | 0.4382 | 0.7887 | 0.3751 | 0.9022 | 0.3093 | 0.9652 | 0.2081 | 0.9828 | 0.1664 |

# H  EFFECT OF MODULUS AMPLIFICATION NETWORK

To study the effectiveness of the proposed modulus amplification network (See Section 3.2), we visualize the representations before and after modulus amplification in the complex plane. The results on the Frappe, ML-Tag, Criteo and Avazu datasets are shown in Figure 8. We can observe that, before the modulus amplification procedure, the feature representations are distributed on a unit circle with a fixed modulus of 1. Specifically, the angular representations learned in RFM vary from $[-\pi, \pi]$ on the ML-Tag, Criteo and Avazu datasets. Whereas on the Frappe datasets, due to its smaller scale, the range is narrowed to $[-\pi/10, \pi/10]$. After amplification, the features are distributed at various areas in the complex plane, and they have different modulus. Specially, we can also see that most transformed representations have the same real part or imaginary part. Such distributions make the varies of angle have a remarkable influence on the predicted result, which enables RFM to capture the useful feature relationships and improves the model's capabilities.

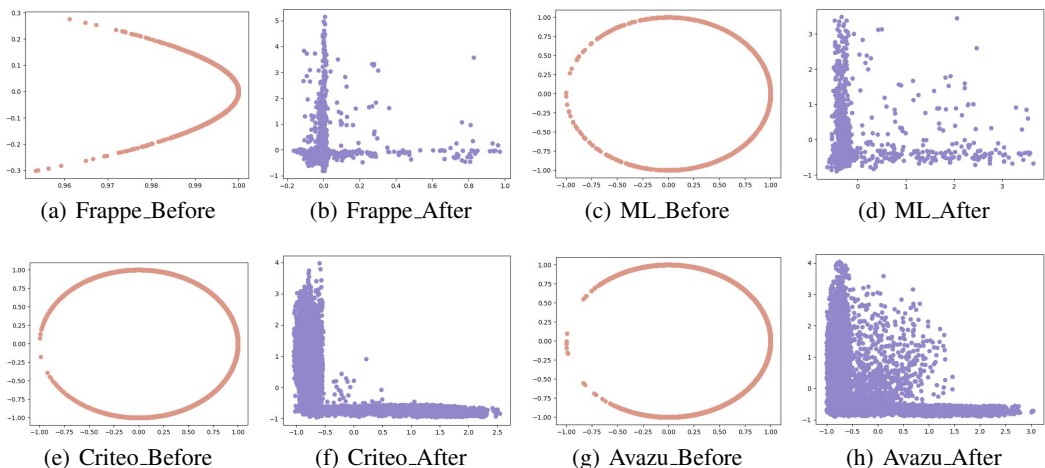

|  (a) Frappe_Before | (b) Frappe_After | (c) ML_Before | (d) ML_After |
|---|---|---|---|
| (e) Criteo_Before | (f) Criteo_After | (g) Avazu_Before | (h) Avazu_After |

Figure 8: Visualization of the feature representations before and after the amplification.

# I  HIGH-ORDER INTERACTION LEARNING ANALYSIS

As discussed in Section 3.3, our proposed method can be degenerated to the traditional inner-product-based methods. To study the effectiveness of the proposed rotation-based interaction in learning high-order feature interactions, we create synthetic datasets with increasing difficulty as:

$$f_m(\boldsymbol{E}) = \boldsymbol{e}_1 \odot \boldsymbol{e}_2 \odot \cdots \odot \boldsymbol{e}_m. \tag{19}$$

where the set $\boldsymbol{E} = \{\boldsymbol{e}_1, \boldsymbol{e}_2, \cdots, \boldsymbol{e}_m\}$, and each $\boldsymbol{e}_j$ is uniformly sampled from [-1, 1]. We compare the prediction result learned in RFM and a complex MLP, and utilize fitting deviation to evaluate the difference between the prediction results of the models and the ground-truth high-order feature interactions (*i.e.,* $f_m$). As shown in Figure 9, we can observe that the fitting deviation continuously decreases as dimensions increase. As the task difficulty increases (the order $m$ increases), the fitting deviation also grows. This is consistent with the theoretical analysis in Section 3.3. On the other hand, the deviation of RFM is very small ($10^{-2}$), which is almost 100 times smaller than it in the Complex MLP model, showing the approximately lossless fitting capability of RFM in learning high-order feature interactions.

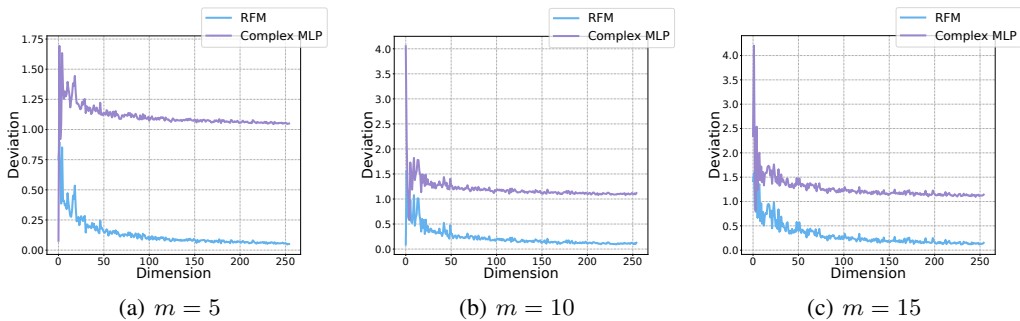

(a) $m = 5$            (b) $m = 10$            (c) $m = 15$

Figure 9: The fitting deviation curves of different learning models.

