# OpenReview forum: "Rotative Factorization Machines"
_ICLR.cc/2024/Conference — Submitted to ICLR 2024_

### Official Review · Reviewer_5eVE · 2023-10-30

**Soundness:** 3 good
**Presentation:** 3 good
**Contribution:** 2 fair
**Rating:** 5
**Confidence:** 4

**Summary:**

This paper introduces a rotative factorization machine method, which represents each feature as a polar angle in the complex plane and converts the feature interactions into a series of complex rotations. The authors design a self-attentive rotation function that models the rotation coefficients through a rotation-based attention mechanism and incorporate a modulus amplification network to learn the modulus of the complex features to enhance the representations. The experiments were conducted on five datasets.

**Strengths:**

1. The proposed method can handle a large number of feature interactions.

2. The paper is well-written and easy to follow.

**Weaknesses:**

The experiment is limited on AUC and LogLoss. How about other machine learning tasks?
In recent machine learning tasks, we are often using AUPRC, instead of AUC.
The improvement in experiments is marginal.

**Questions:**

Please check the weakness section.

---

> ### Author Response · Authors · 2023-11-11
> **Response to the Concerns of the Reviewer**
>
> We sincerely appreciate your insightful comments and the opportunity to address your concerns regarding the evaluation metrics employed in our study. I would like to provide a thoughtful response. If you still have any other questions, please do not hesitate to inform us. We will continue to do our best to provide answers for you.
>
> #### **The experiment is limited to AUC and LogLoss. How about other machine learning tasks? In recent machine learning tasks, AUPRC is often used instead of AUC.**
>
> In the realm of feature interaction learning, particularly in the context of ranking tasks such as CTR prediction and product recommendation, the use of AUC and LogLoss has become a standard evaluation protocol across a large number of studies [1][2][3][4][5][6][7]. Specifically, AUC is widely accepted for its effectiveness in evaluating the ranking performance of models, especially in scenarios where ranking plays a crucial role. LogLoss, on the other hand, provides valuable insights into the probabilistic predictions of the model. For consistency and comparability within this field, we follow most of the previous works [1][2][3][4][5][6][7] that adopt these metrics as a benchmark for evaluating the model's performance.
>
> By aligning with this established protocol, our aim is to ensure methodological rigor and facilitate meaningful comparisons with existing literature, especially with those closely related studies [1][2][3]. Therefore, following previous research efforts, we only tested the most commonly used machine learning tasks on the widely used datasets for feature interaction approaches. In the future, we will try to apply our approach to other scenarios or tasks. We hope that our work can make a broader contribution to the research community.
>
> #### **The improvement in experiments is marginal.**
> In the research field of feature interaction learning, we often adopt AUC and LogLoss as metrics to evaluate model performance. Previous studies [1][2][3][4][5][6][7] have stated that a higher AUC or a lower LogLoss at the 0.001 level is regarded as significant. Specifically, a prior study [7] has pointed out that a slightly higher offline AUC is more significant in online traffic. In particular, our experimental results and metrics remain consistent with the majority of previous works [1][2][4][5], and the results indicate that the improvement achieved by our method is significant.
> We utilized an open-source code repository and released our code, executed multiple runs for each result, and conducted significance tests, all of which can demonstrate the effectiveness of our proposed approach and ensure its reproducibility.
>
> [1] EulerNet: Adaptive Feature Interaction Learning via Euler’s Formula for CTR Prediction, SIGIR 2023.
>
> [2] Adaptive Factorization Network: Learning Adaptive-Order Feature Interactions, AAAI 2020.
>
> [3] ARM-Net: Adaptive Relation Modeling Network for Structured Data, SIGMOD 2021.
>
> [4] DCN V2: Improved Deep & Cross Network and Practical Lessons for Web-scale Learning to Rank Systems, WWW 2021.
>
> [5] 𝐹𝑀 $^2$: Field-matrixed Factorization Machines for Recommender Systems, WWW 2021.
>
> [6] AutoInt: Automatic Feature Interaction Learning via Self-Attentive Neural Networks, CIKM 2019.
>
> [7] Wide & Deep Learning for Recommender Systems, DLRS 2016.

---

### Official Review · Reviewer_LHqT · 2023-10-31

**Soundness:** 2 fair
**Presentation:** 3 good
**Contribution:** 2 fair
**Rating:** 5
**Confidence:** 5

**Summary:**

This paper proposes a Rotative Factorization Machine (RFM), where feature interactions are converted into a series of complex rotations to facilitate arbitrarily-order feature interaction learning.

**Strengths:**

Learning arbitrary order feature interaction with complex rotations is interesting.

**Weaknesses:**

- The proposed method is similar to EulerNet with a stack of attention module.
- The Theorem is not properly defined, used and explained.
- There is no detail about dataset train/test split. The paper refers to another paper which refers to another paper where I still could not find clear defined split. Benchmarking is not rigorous and is hard to tell whether the proposed method is better or not in terms of performance, especially there are given prior work for discussion [1,2]. Apart from that, the improvement on ML-1M, ML-Tag, Frappe is marginal, and from my experience the improvement could be a result of randomness.

[1] Jieming Zhu, Jinyang Liu, Shuai Yang, Qi Zhang, Xiuqiang He. Open Benchmarking for Click-Through Rate Prediction. The 30th ACM International Conference on Information and Knowledge Management (CIKM), 2021.

[2] Jieming Zhu, Quanyu Dai, Liangcai Su, Rong Ma, Jinyang Liu, Guohao Cai, Xi Xiao, Rui Zhang. BARS: Towards Open Benchmarking for Recommender Systems. The 45th International ACM SIGIR Conference on Research and Development in Information Retrieval (SIGIR), 2022.

**Questions:**

In addition to the weakness above, I have following questions:
1) "To our knowledge, it is the first work that is capable of learning the interactions with arbitrarily large order adaptively from the corresponding interaction contexts." Isn't this also done by [1]?
2) What is the training time for the proposed model? How does it compare to other models?
3) How is the theorem useful/helpful? Since it gives an asymptotic property of $\Delta_{RFM}$ to $\Delta_{R}$, but
 - There is no reason for $\Delta_{R}$ to be better than $\Delta_{RFM}$, why we approximate it?
 - If $\Delta_{R}$ is indeed better, why not directly use $\Delta_{R}$ to capture feature interaction?
 - Lemma A.1 holds when you assume the embeddings are random vectors. They however are not.

4) Can the proposed method be interpreted as using dual embedding vectors for one feature, with the constraint that they have unit norm in one dimension?


[1] Weiyu Cheng, Yanyan Shen, and Linpeng Huang. Adaptive factorization network: Learning adaptive-order feature interactions. In Proceedings of the AAAI Conference on Artificial Intel- ligence, volume 34, pp. 3609–3616, 2020.


==================================

Thanks for the response. I tend to keep my score because:

1. the model still is too similar to EulerNet, despite it replaces the linear block with an attention block. The claims regarding arbitrarily large order and exponential explosion issue are not supported. It could be redundant to separately learn modulus and phase. If exponential explosion issue occurs, I couldn't see how such scheme could help. The modulus could still be too large;

2. the Theorem does not support the model well. A simple MLP would do the same as a universal approximator, without all these loose/inappropriate bounds, under which I think related work could be as powerful as the proposed method.

---

> ### Author Response · Authors · 2023-11-12
> **Response to the Concerns of the Reviewer [Part 1]**
>
> Thanks for your insightful suggestions and we have listed our response to your concerns as follows. If you also have any other questions, please feel free to let us know. We will continue to try our best to answer for you.
>
> #### **1. The proposed method is similar to EulerNet with a stack of attention module.**
> In Section 3.3, we highlight the novelty and differences between our approach and other feature interaction methods.
> Our proposed method and EulerNet [1] both model the feature interactions in the complex vector space, but there are still some noticeable differences:
>
> - First, EulerNet [1] uses simple **linear layers** to model arbitrary-order  feature interactions, with interaction orders determined by predefined linear combination coefficients.
> In contrast, our approach develops a **self-attentive rotation mechanism** to model feature interactions, with interaction orders learned from attention scores derived from the context.
> This enables our approach to effectively capture feature dependencies in varying contexts.
>
> - Second, EulerNet [1] simultaneously optimizes the modulus and phase of the complex feature interactions in a coupled way, leading to the the **exponential explosion** issue when dealing with a large order, due to the exponential growth in the modulus of complex features.
> In contrast, we decouple the learning processes of the modulus and phase of complex features. Specifically, we use complex rotations to model complicated interactions and incorporate a modulus amplification network to enhance representations.
> This approach effectively avoids the exponential explosion issue, allowing for the learning of **arbitrarily large order**.
>
> [1] EulerNet: Adaptive Feature Interaction Learning via Euler’s Formula for CTR Prediction, SIGIR 2023.
>
> #### **2. There is no detail about dataset train/test split. Benchmarking is not rigorous and is hard to tell whether the proposed method is better or not in terms of performance.**
> To maintain consistency with previous work, we follow the AFN [1] to process the ML-Tag and Frappe dataset, where the split ratio for the train/val/test is 7:2:1; we also follow the EulerNet [2] to process the Criteo, Avazu and ML-1M dataset, where the split ratio is 8:1:1.
> Previous studies [1][2][3][4][5] have stated that a higher AUC or a lower LogLoss at the **0.001** level is regarded as significant.
> More details can be found in their code repository.
> In particular, our experimental results and metrics remain consistent with the majority of previous works [1][2][3][4][5], and the results indicate that the improvement achieved by our method is significant.
> We utilized an open-source code repository and released our code, executed multiple runs for each result, and conducted significance tests, all of which can demonstrate the effectiveness of our proposed approach and ensure its reproducibility.
>
> [1] Adaptive Factorization Network: Learning Adaptive-Order Feature Interactions, AAAI 2020.
>
> [2] EulerNet: Adaptive Feature Interaction Learning via Euler’s Formula for CTR Prediction, SIGIR 2023.
>
> [3] DCN V2: Improved Deep & Cross Network and Practical Lessons for Web-scale Learning to Rank Systems, WWW 2021.
>
> [4] 𝐹𝑀 $^2$: Field-matrixed Factorization Machines for Recommender Systems, WWW 2021.
>
> [5] Towards Deeper, Lighter and Interpretable Cross Network for CTR Prediction, CIKM 2023.

---

> ### Author Response · Authors · 2023-11-12
> **Response to the Concerns of the Reviewer [Part 2]**
>
> #### **3. "To our knowledge, it is the first work that is capable of learning the interactions with arbitrarily large order adaptively from the corresponding interaction contexts." Isn't this also done by AFN [1]?**
> Although AFN [1] can adaptively learn the feature interaction orders to some extent, but:
> - First, the order of feature interactions in AFN [1] cannot take arbitrarily large values, due to the exponential explosion issue, e.g., considering the interaction term $\boldsymbol e_j^{\alpha_j}$ with embedding $\boldsymbol e_j = [0.5, 2]$, its order $\alpha_j$ cannot be 100, since $2^{100}$ is exceptionally large, which makes them suffer from the gradient explosion issue.
> We have also provided the detailed analysis in the Section 4.3 (**Arbitrary-Order Learning Analysis**).
> Such approaches  cannot scale to high-order cases in industrial scenarios.
> As our solution, we project features to the complex plane (polar angles) that converts the interactions into complex rotations.
> Such transformation can effectively avoid the exponential explosion issue and allows for the learning of **arbitrary large order**. (The detailed proofs are also provided in Section 3.1.)
>
> - Second, the orders learned in AFN [1] cannot capture the dependencies between different features, i.e., the interaction order of each feature is independently learned.
> However, as prior work [2] shows, in real-world applications, the importance  of a certain feature (controlled by the order) is often influenced by other features.
> It is challenging for AFN [1] to effectively capture the varied feature importance in different interaction contexts.
> Unlike AFN, the interaction orders learned in our approach can effectively capture the dependencies between different features through the proposed self-attentive rotations, enabling it to learn the varied interaction orders from **the corresponding interaction contexts**.
>
> [1] Adaptive Factorization Network: Learning Adaptive-Order Feature Interactions, AAAI 2020.
>
> [2] Enhancing CTR Prediction with Context-Aware Feature
> Representation Learning, SIGIR 2022.
>
> #### **4. What is the training time for the proposed model? How does it compare to other models?**
>
> We have discussed the inference efficiency of the model in Table 3 and analyzed the model complexity in Appendix E.
> Our model's overall complexity aligns with that of the transformer. To gain a more comprehensive understanding of the complexity information of the model, we present the training times for the experiments in the table below.
> Please note that, due to the one-epoch phenomenon [1], we provide the training time (in seconds) for one epoch to facilitate a fair comparison.
>
> | Model | Criteo | Avazu | ML-1M | ML-Tag | Frappe |
> |----------|----------|----------|----------|----------|----------|
> |   LR  |   355.51  |   252.98  |   2.42  |   4.30  |   0.76  |
> |   FwFM  |   375.87  |   301.37  |   2.57  |   4.71  |   1.03  |
> |   FmFM  |   610.29  |  530.49  |   2.99  |   6.94  |   1.26  |
> |   NFM  |   565.35  |   464.63  |   5.07 |   10.30  |   2.76  |
> |   CIN  |   803.92  |   713.59  |   9.93  |   15.51  |   3.51  |
> |   CrossNet  |   452.21  |   403.73  |   3.39  |   11.47  |   1.37  |
> |   PNN  |   654.77  |   332.21  |  4.84  |   10.65  |   1.60  |
> |   Transformer  |  745.99   |   563.52  |   8.08  |   21.19  |   3.74  |
> |   DeepFM  |   445.21  |  384.64  |   4.50  |   10.49  |  1.63  |
> |  xDeepFM  |  857.64  |  724.78  |  10.94  |  17.77  |  3.84  |
> |  DCNV2  |  474.63  |  410.39  | 3.70  |  12.57  |  1.45  |
> |  AFN+  |  930.74  |  745.45  |  10.03  |  16.73  |  21.72 |
> |  ARM-Net+  |  1754.36  |  1140.30  |  14.89 |  30.69  |  27.94  |
> |  EulerNet  |  597.96  |  510.22  |  5.29  |  11.09  |  1.82  |
> |  RFM  |  783.43  |  681.27  |  6.11  |  17.59  |  1.92  |
>
> [1] Towards Understanding the Overfitting Phenomenon of Deep Click-Through Rate Prediction Models, CIKM 2022.

---

> ### Author Response · Authors · 2023-11-12
> **Response to the Concerns of the Reviewer [Part 3]**
>
> #### **5. The Theorem is not properly defined, used and explained. How is the theorem useful/helpful? Since it gives an asymptotic property of $\Delta_{RFM}$ to $\Delta_{R}$, but:**
>
> **(1) There is no reason for $\Delta_{R}$ to be better than $\Delta_{RFM}$, why we approximate it?**
>
> - We approximate $\Delta_{R}$ to demonstrate that our method (**rotation-based** interaction) can be instantiated as any traditional **inner-product-based** feature interaction model (e.g. FM [1], EulerNet [2], AFN [3]), since the interaction learning function of arbitrary traditional models [1][2][3] can be expressed using $\Delta_{R}$. Therefore, any traditional model can be viewed as a special case of our proposed method, demonstrating the effectiveness of our approach.
>
> **(2) If $\Delta_{R}$
>  is indeed better, why not directly use $\Delta_{R}$
>  to capture feature interaction?**
>  - Previous studies [1][2][3] have demonstrated the effectiveness of using $\Delta_{R}$ to model feature interactions. However, as mentioned earlier, **$\Delta_{R}$ cannot effectively capture arbitrary high-order feature interactions** (due to the exponential explosion issue).
>  In contrast, our approach ($\Delta_{RFM}$) not only has approximate results to $\Delta_{R}$ when the order is low, but also can scale to **arbitrarily high orders** of feature interactions, thereby illustrating the effectiveness and superiority of our proposed approach.
>
>  **(3) Lemma A.1 holds when you assume the embeddings are random vectors. They however are not.**
>
> - As a recent study [4] shows, the feature embedding space can be regarded as an unobservable random variable sampled from a certain distribution.
> Following the study [4], on the strength of Mean-field Theory [5], we suppose different fields are mutually independent.
> Thus, under the constraint of the L2 norm, for ease of theoretical analysis, we approximate it by sampling within a unit hypersphere.
> Because for an unknown machine learning task, we cannot specify a particular and accurate distribution; it may be distributed in any region of the unit sphere.
>
>  **(4) Can the proposed method be interpreted as using dual embedding vectors for one feature, with the constraint that they have unit norm in one dimension?**
>
> - Yes, this can be understood from a certain perspective.
> As we represent a feature using a vector located on the unit circle in the complex plane, this complex vector can be expressed using two real vectors, i.e., $\mathbf r + i \mathbf p$, s.t., $\mathbf r^2 +  \mathbf p^2 = \mathbf 1$.
> Specifically, we use the arctangent function ($atan2$) of these dual embedding vectors to obtain the polar angle vectors in the complex plane, and model the feature interactions through the rotation transformations based on these polar vectors.
>
> [1] Factorization Machines, ICDM 2010.
>
> [2] EulerNet: Adaptive Feature Interaction Learning via Euler’s Formula for CTR Prediction, SIGIR 2023.
>
> [3] Adaptive Factorization Network: Learning Adaptive-Order Feature Interactions, AAAI 2020.
>
> [4] Alleviating Cold-start Problem in CTR Prediction with A
> Variational Embedding Learning Framework, WWW 2022.
>
> [5] Variational inference: A review for statisticians, JASA 2017.

---

### Official Review · Reviewer_f3tV · 2023-11-01

**Soundness:** 4 excellent
**Presentation:** 3 good
**Contribution:** 3 good
**Rating:** 6
**Confidence:** 5

**Summary:**

This paper present the RFM  to learn arbitrary orders for the CTR task. Mathematical proofs, extensive experimental results are provided to show the properties and advantages of this work.

**Strengths:**

This is a novel, complete, well-presented paper. Specifically, it formulates the feature interaction from a new perspective, representing features as a polar angle in the complex plane. Extensive experiments verify the effectiveness of RFM on 5 datasets. Source code is also provided.

**Weaknesses:**

1. Some of the concepts are different from the existing CTR papers, will be good to be consistent or make it clearer. For example, the field-aware means each feature has multiple field-aware representations [1]. The term "relation-aware" is more like "feature-aware order" to me, which indicates the order information a is depends on the indicator of input features. This is something similar to the second-order feature interaction in the existing papers. In other words, the feature interaction information in the existing papers is treated as order information in this paper. Compared with AFN [2], the order information in AFN is not explicitly modled.

[1] Field-aware Factorization Machines for CTR Prediction, RecSys 16.
[2] Adaptive Factorization Network: Learning Adaptive-Order Feature Interactions, AAAI 2020

2. The motivation of the projecting features to the complex plane is not clear. Take AFN [2] as an example, it has similar capability to model arbitrary orders. What the diffidence between the existing solutions (e.g., Logarithmic Transformation) and yours should be discussed.

**Questions:**

1. Is equation 3 a learning function for 1 feature interaction? Based on the equation and figure 2, it looks like they are an illustration on how to learn a specific feature interaction in the context of feature j. Because only the pairwise relationships between feature j and other features are modeled  (i.e., x_j and x_1, x_j and x_l, and x_j and x_m), which are represented as a_{j,i}, a_{j,l}, and a_{j,m} respectively.
How to model the relationship a_{1,m} between x_1 and x_m in this case? How to indicate the number of feature interactions?
In my opinion, you might be referring to the Equ 7 in  AFN [1], which is the output of the jth neuron. If yes, how to define the number of neurons (i.e., feature interactions)?

[1] Adaptive Factorization Network: Learning Adaptive-Order Feature Interactions, AAAI 2020.

I would like to increase the rating if the authors can solve my concerns. Thanks for the great efforts, I enjoy reading this paper!

**Details Of Ethics Concerns:**

N.A.

---

> ### Author Response · Authors · 2023-11-11
> **Response to the Concerns of the Reviewer [Part 1]**
>
> Thanks for your insightful suggestions and we have listed our response to your concerns as follows. If you also have any other questions, please feel free to let us know. We will continue to try our best to answer for you.
>
> #### **For weakness (1):**
> We sincerely appreciate the terminology questions raised by the reviewer. In the field of click-through rate (CTR) predictions, the term **field-aware** was primarily introduced by the work of FFMs [1]. It mainly refers to multi-field feature embeddings and has subsequently spawned many other related concepts.
> For example, in the context of neural architecture search [2], **field-aware** refers to searching the optimal embedding dimension at the **field-level**.
> In this paper, we use the terms (**field-aware**, **instance-aware** and **relation-aware**) to denote the learning granularities of interaction orders, in consideration of maintaining consistency with the term **instance-aware** proposed in ARM-Net [3], a closely related work.
>
> Specifically, we explained the concept of different learning granularities of interaction orders in Fig. 1.
> For ease of illustration, we present the interactions with only two fields.
> Specially, AFN [4] and EulerNet [5] simply assign a shared order for all features within each field, and thus, the interaction order is only aware of **field-level** information.
> Therefore, we refer to them as **field-aware** models.
> As for ARM-Net [3] (**instance-aware**), the interaction order is learned based on the indicator of input features; i.e., each feature instance in a given interaction is assigned with a unique order, which is aware of instance-level information.
> As for our approach (**relation-aware**), the feature interaction order is explicitly modeled through an attention mechanism; i.e., each feature combination (query-key pair) in a given interaction is assigned with a unique order (denoted as $\alpha_{j,l}$), which is aware of the pairwise feature **relationships** in the context of any given feature $j$.
>
>
>
> #### **For weakness (2):**
> In Section 3.1, we have discussed the main shortcomings of existing methods (e.g., AFN [4]) and the motivation behind our approach:
>
> - First, the order of feature interactions in AFN [4] cannot take arbitrarily large values, due to the exponential explosion issue, e.g., considering the interaction term $\boldsymbol e_j^{\alpha_j}$ with embedding $\boldsymbol e_j = [0.5, 2]$, its order $\alpha_j$ cannot be 100, since $2^{100}$ is exceptionally large, which makes them suffer from the gradient explosion issue.
> We have also provided the detailed analysis in the Section 4.3 (**Arbitrary-Order Learning Analysis**).
> Such approaches  cannot scale to high-order cases in industrial scenarios.
> As our solution, we project features to the complex plane (polar angles) that converts the interactions into complex rotations.
> Such transformation can effectively avoid the exponential explosion issue and allows for the learning of arbitrary large order. (The detailed proofs are also provided in Section 3.1.)
>
> - Second, the orders learned in field-aware and instance-aware methods cannot capture the dependencies between different features, i.e., the interaction order of each feature is independently learned.
> However, as prior work [6] shows, in real-world applications, the importance  of a certain feature (controlled by the order) is often influenced by other features.
> It is challenging for both
> field-aware and instance-aware models to effectively capture the varied feature importance in different interaction contexts.
> Unlike them, the interaction orders learned in our approach can effectively capture the dependencies between different features through the proposed self-attentive rotations.
>
> [1] Field-aware Factorization Machines for CTR Prediction, RecSys 2016.
>
> [2] AutoDim: Field-aware Embedding Dimension Search in Recommender Systems, WWW 2021.
>
> [3] ARM-Net: Adaptive Relation Modeling Network for Structured Data, SIGMOD 2021.
>
> [4] Adaptive Factorization Network: Learning Adaptive-Order Feature Interactions, AAAI 2020.
>
> [5] EulerNet: Adaptive Feature Interaction Learning via Euler’s Formula for CTR Prediction, SIGIR 2023.
>
> [6] Enhancing CTR Prediction with Context-Aware Feature
> Representation Learning, SIGIR 2022.

---

> ### Author Response · Authors · 2023-11-11
> **Response to the Concerns of the Reviewer [Part 2]**
>
> #### **List of questions.**
>
> (a) Is equation 3 a learning function for 1 feature interaction?
>
> (b) How to model the relationship a_{1,m} between x_1 and x_m in this case?
>
> (c) How to indicate the number of feature interactions?
>
> (d) In my opinion, you might be referring to the Equ 7 in AFN [1], which is the output of the jth neuron. If yes, how to define the number of neurons (i.e., feature interactions)?
>
> - Answer (a): the equation (3) describes the interaction learning function with multiple interaction terms, and the set of order vectors is denoted as $\mathcal{A}$. For example, considering the interaction $\tilde{\boldsymbol e}_1 \odot \tilde{\boldsymbol e}_2 + \tilde{\boldsymbol e}_2 \odot \tilde{\boldsymbol e}_3 + \tilde{\boldsymbol e}_3 \odot \tilde{\boldsymbol e}_3$ with three fields ($m=3$), the set of order vectors is $\mathcal{A} = {[1,1,0], [0,1,1],[1,0,1]}$. (Specifically, $[1,1,0]$ for $\tilde{\boldsymbol e}_1 \odot \tilde{\boldsymbol e}_2$, $[0,1,1]$ for $\tilde{\boldsymbol e}_2 \odot \tilde{\boldsymbol e}_3$, and $[1,0,1]$ for $\tilde{\boldsymbol e}_1 \odot \tilde{\boldsymbol e}_3$) Therefore, each order vector corresponds to an interaction term, and the number of feature interaction terms is the number of order vectors.
>
> - Answer (b): The feature interaction order $\alpha_{1,m}$ is modeled by the attention score between the query $\boldsymbol Q_1$ and the key $\boldsymbol K_{m}$. In our approach, we use self-attentive rotations to model relation-aware feature interactions. Specifically, we have $m$ queries ($\boldsymbol Q_1, \boldsymbol Q_2, ..., \boldsymbol Q_m$) and $m$ keys ($\boldsymbol K_1, \boldsymbol K_2, ..., \boldsymbol K_m$), and each query (e.g., $\boldsymbol Q_j$) will match all keys ($\boldsymbol K_1, \boldsymbol K_2, ..., \boldsymbol K_m$), generating $m$ interaction orders ($\alpha_{j,1}, \alpha_{j,2}, ..., \alpha_{j,m}$), constructing an interaction term $\tilde e_1^{\alpha_j,1} \odot \tilde e_2^{\alpha_j,2} \odot \cdots \odot\tilde e_m^{\alpha_j,m}$., measuring the relationships in the context of feature $j$.
> Therefore, the relationship $\alpha_{1,m}$ is modeled by the first query $\boldsymbol Q_1$ with the $m$-th Key $\boldsymbol K_{m}$.
>
> - Answer (c): as mentioned in Answer (b), each query will construct a feature interaction term $\tilde{\boldsymbol e}_1^{\alpha_j,1} \odot \tilde{\boldsymbol e}_2^{\alpha_j,2} \odot \cdots \odot \tilde{\boldsymbol e}_m^{\alpha_j,m}$.
> Therefore, the number of feature interactions is $m$ for a single head attention.
> As introduced in Section 3.1.2 (**Multi-Head Rotation**), we use the head number $h$ to control the number of feature interaction terms (thus the total term number is $mh$).
> All the output representations are concatenated as the input of the MLP, which enables it to filter out the useless interaction terms.
>
> - Answer (d): As introduced in Section 3.1.2, a feature interaction term (the Equ 7 in AFN [1], output of the j-th neuron) is also described in Eq. (7) in our paper ($\mathcal{F}(\mathcal{A}_j):=\tilde{\boldsymbol e}_1^{\alpha_j,1} \odot \tilde{\boldsymbol e}_2^{\alpha_j,2} \odot \cdots \odot \tilde{\boldsymbol e}_m^{\alpha_j,m}$), where the order is generated by the attetnion score from $j$-th query.
> Therefore, the number of feature interactions (neurons in AFN [1]) is the number of queries (i.e., $m$) for a single-head in our approach, and each query corresponds to a neuron.
> For $h$ attention heads, the number of feature interactions is $mh$.
>
> [1] Adaptive Factorization Network: Learning Adaptive-Order Feature Interactions, AAAI 2020.

---

> ### Comment · Reviewer_f3tV · 2023-11-23
>
> Thanks for considering my comments and the detailed replies for my questions, I will slightly adjust my score.

---

### Official Review · Reviewer_7fiD · 2023-11-02

**Soundness:** 2 fair
**Presentation:** 2 fair
**Contribution:** 2 fair
**Rating:** 3
**Confidence:** 2

**Summary:**

This paper proposes a new model, named RFM (Rotative Factorization Machine), that can capture higher order feature interactions. Instead of directly using feature embeddings, RFM applies exponential operator over the embedings, called angular representation of features. With that, the cross of features (say e^x*e^y*e^z) becomes the sum of transformed features (say, e^{x+y+z}). The authors employs transformer-like self-attention structures in RFM. RFM comes with a subnetwork which does modulus amplification for better capture of complex feature interactions. The authors conduct experiments over several datasets and showed RFM beats other state-of-the-art models.

**Strengths:**

* RFM has some good features for capturing feature interaction. It can compute arbitrary high-order feature interactions with help of the angular representation of features. It models feature interaction with rotating vectors in a high dimensional space.
* From the empirical evaluation results, the modulus amplification subnetwork does help RFM's overall performance a lot.

**Weaknesses:**

* RFM is positioned as a better model for capturing arbitrary high order complex feature interactions. A complex MLP can also do that. One key question is whether the proposed model is good at capturing high-order complex features, and how it achieves that. I don't really see too many discussions or analysis over this direction. In the paper's current form, I doubt readers would be convinced that those hypothetical complex vector rotations are the keys to capture feature interactions better.
* Section 4 doesn't involve enough evidence that RFM is capturing feature interactions more efficiently, other than showing it has better prediction performance.

**Questions:**

* See my comments in the weakness section. I wonder if there could be more supporting discussions/materials that show how RFM's angular representations/rotation-based attention/modulus amplification are helping to better capture complex feature interactions. One thing that might help to convince the readers is to generate some synthetic data to show complex feature interactions are degenerated to simpler ones after these transformations.
* Question regarding the angular representations. Theoretically, the embedding layers could learn the exponential transformations. Say instead of x_j --> \theta_j (=E(x_j)) --> {\tilde e}_j (=e^{i\theta_j}, there could be a different embedding (\hat E) such that \hat E(x_j) = e^{iE(x_j)}. Why does explicitly adding this exponential transformation help? What's the hidden cost of it?

---

> ### Author Response · Authors · 2023-11-12
> **Response to the Concerns of the Reviewer [Part 1]**
>
> Thanks for your insightful suggestions and we have listed our response to your concerns as follows. If you also have any other questions, please feel free to let us know. We will continue to try our best to answer for you.
>
>  **1. One key question is whether the proposed model is good at capturing high-order complex features, and how it achieves that.**
>
> As discussed in Section 3.1, arbitrary-order feature interactions can be converted into the complex rotations in our approach.
>
> - In the theoretical aspect, we have proven in Section 3.3 that our rotation-based interactions can achieve the similar prediction effect to traditional inner-product high-order feature interactions, i.e., our proposed complex feature interactions can be degenerated to simpler ones after these transformations.
>
>
> - In terms of experiments, we have explored the model's ability to learn arbitrary higher-order feature interactions in Section 4.3 (**Arbitrary-Order Learning Analysis**).
> Meanwhile, the model performance presented in Table 3 further illustrate the effectiveness of our proposed approach (better than Complex MLP, i.e., EulerNet).
> Further, in the ablation study (Section 4.3), after we remove the rotation component, the model performance shows a significant decrease, indicating that the proposed rotations are the keys for learning effective feature interactions.
> **To verify the degree of coincidence with the high-order feature interactions learned in RFM, we add the experiments using synthetic
> data in Appendix I.**
>
>
>
>
> **2. Section 4 doesn't involve enough evidence that RFM is capturing feature interactions more efficiently, other than showing it has better prediction performance.**
>
> **We have discussed the inference efficiency of the model in Table 3 and analyzed the model complexity in Appendix E**.
> Our model's overall complexity aligns with that of the transformer. To gain a more comprehensive understanding of the complexity information of the model, we present the training times for the experiments in the table below.
> Please note that, due to the one-epoch phenomenon [1], we provide the training time (in seconds) for one epoch to facilitate a fair comparison.
>
> | Model | Criteo | Avazu | ML-1M | ML-Tag | Frappe |
> |----------|----------|----------|----------|----------|----------|
> |   LR  |   355.51  |   252.98  |   2.42  |   4.30  |   0.76  |
> |   FwFM  |   375.87  |   301.37  |   2.57  |   4.71  |   1.03  |
> |   FmFM  |   610.29  |  530.49  |   2.99  |   6.94  |   1.26  |
> |   NFM  |   565.35  |   464.63  |   5.07 |   10.30  |   2.76  |
> |   CIN  |   803.92  |   713.59  |   9.93  |   15.51  |   3.51  |
> |   CrossNet  |   452.21  |   403.73  |   3.39  |   11.47  |   1.37  |
> |   PNN  |   654.77  |   332.21  |  4.84  |   10.65  |   1.60  |
> |   Transformer  |  745.99   |   563.52  |   8.08  |   21.19  |   3.74  |
> |   DeepFM  |   445.21  |  384.64  |   4.50  |   10.49  |  1.63  |
> |  xDeepFM  |  857.64  |  724.78  |  10.94  |  17.77  |  3.84  |
> |  DCNV2  |  474.63  |  410.39  | 3.70  |  12.57  |  1.45  |
> |  AFN+  |  930.74  |  745.45  |  10.03  |  16.73  |  21.72 |
> |  ARM-Net+  |  1754.36  |  1140.30  |  14.89 |  30.69  |  27.94  |
> |  EulerNet  |  597.96  |  510.22  |  5.29  |  11.09  |  1.82  |
> |  RFM  |  783.43  |  681.27  |  6.11  |  17.59  |  1.92  |
>
> [1] Towards Understanding the Overfitting Phenomenon of Deep Click-Through Rate Prediction Models, CIKM 2022.
>
>  **3. One thing that might help to convince the readers is to generate some synthetic data to show complex feature interactions are degenerated to simpler ones after these transformations.**
>
> Thanks for your careful reading of our paper. We have tried our best to elaborate the unclear points (High-order feature interaction learning analysis) and revised our paper accordingly. We have added the experiments using synthetic
> data in Appendix I.
> We can see that our approach can degenerated to simpler ones (traditional inner-product based interactions) after these transformations, which is consistent with our theoretical analysis.

---

> ### Author Response · Authors · 2023-11-12
> **Response to the Concerns of the Reviewer [Part 2]**
>
> **4. Question regarding the angular representations. Theoretically, the embedding layers could learn the exponential transformations. Say instead of x_j --> \theta_j (=E(x_j)) --> {\tilde e}_j (=e^{i\theta_j}, there could be a different embedding (\hat E) such that \hat E(x_j) = e^{iE(x_j)}. Why does explicitly adding this exponential transformation help? What's the hidden cost of it?**
>
>  We greatly appreciate the reviewer for raising this question. The reason we use this form is primarily to explicitly emphasize that the feature embeddings we traditionally obtain correspond to the polar angles on the complex plane.
>  Based on this concept, we can subsequently use a rotational mechanism to represent feature interactions.
>
>  In other words, this is merely a conceptual shift rather than an actual transformation.
>  In the computer memory or practice implementation, we are still using the variable $E(\mathbf x_j)$ rather than $e^{iE(\mathbf x_j)}$, and the subsequent attention mechanism is designed for a series of rotation transformations based on $E(\mathbf x_j)$ (also $\mathbf \theta_j$).
>  Based on this concept, after rotation, we use coordinate transformation (Eq. 12) to obtain the real and imaginary parts of the complex features, according to Euler's formula.
>
>  In conclusion, it is merely an expression of an abstract concept rather than an actual transformation, hence there is no hidden cost.

---

### Meta-Review · Area_Chair_8kLX · 2023-12-06

**Metareview:**

This paper proposes Rotative Factorization Machine, a form of factorization machine that models features as angles and feature interactions as rotations on the complex plane. Some theoretical and experimental results are provided to justify the proposed model.

**Justification For Why Not Higher Score:**

The reviewers have a almost unanimous assessment that the paper has not yet met the acceptance threshold (one reviewer raised the score from "marginal reject" to "marginal accept", but not enough to sway the majority opinion). Major concerns are on the validity of the theorem, its similarity to existing models, and limited improvement on real data.

**Justification For Why Not Lower Score:**

N/A

---

### Decision · Program_Chairs · 2024-01-16

Reject